# Stereoisomeric engineering of aggregation-induced emission photosensitizers towards fungal killing

Wenping Zhu[1,5], Ying Li [2,5], Shaoxun Guo[1], Wu-Jie Guo[1], Tuokai Peng[1], Hui Li [1], Bin Liu[1], Hui-Qing Peng [1,3] ✉ & Ben Zhong Tang [4] ✉

Fungal infection poses and increased risk to human health. Photodynamic therapy (PDT) as an alternative antifungal approach garners much interest due to its minimal side effects and negligible antifungal drug resistance. Herein, we develop stereoisomeric photosensitizers ((Z)- and (E)-TPE-EPy) by harnessing different spatial configurations of one molecule. They possess aggregation-induced emission characteristics and ROS, viz. $^1O_2$ and $O_2^{-\cdot}$ generation capabilities that enable image-guided PDT. Also, the cationization of the photosensitizers realizes the targeting of fungal mitochondria for antifungal PDT killing. Particularly, stereoisomeric engineering assisted by supramolecular assembly leads to enhanced fluorescence intensity and ROS generation efficiency of the stereoisomers due to the excited state energy flow from non-radiative decay to the fluorescence pathway and intersystem (ISC) process. As a result, the supramolecular assemblies based on (Z)- and (E)-TPE-EPy show dramatically lowered dark toxicity without sacrificing their significant photo-toxicity in the photodynamic antifungal experiments. This study is a demonstration of stereoisomeric engineering of aggregation-induced emission photosensitizers based on (Z)- and (E)-configurations.

Fungal infections have greatly threatened human health due to their high morbidity and mortality rate[1–5]. Members of azoles (e.g., fluconazole) are used as the frontline treatment for invasive fungal disease[6]. However, their broad clinical use has synchronously caused unprecedented antifungal resistance. Therefore, developing alternative antifungal therapeutics is urgent and essential for reducing this global burden of public health issues. Photodynamic therapy (PDT) has garnered much interest in treating fungal infections since it is a minimally invasive approach with precise spatiotemporal control, minimal side effects, and negligible drug resistance[7–12]. Its therapeutic efficacy is highly sensitive to molecular structures and intermolecular interactions of photosensitizers (PSs) that affect the light-induced generation of destructive singlet oxygen ($^1O_2$) or other reactive oxygen species (ROS) from molecular oxygen. For example, most conventional PSs possess planar structures, thereby leading to aggregation-caused fluorescence quenching and insufficient ROS generation because of the aromatic π–π stacking in the aqueous environment[13]. In contrast, luminogens with aggregation-induced emission (AIE) characteristics have twisted structures, which in aggregates can hamper the excited state energy dissipation by restricting intramolecular motions, resulting in enhanced fluorescence intensity and efficient ROS generation for image-guided PDT[14]. For this reason, various AIE

[1]Beijing Advanced Innovation Center for Soft Matter Science and Engineering, State Key Laboratory of Chemical Resource Engineering, Beijing University of Chemical Technology, Beijing 100029, China. [2]Guangzhou Municipal and Guangdong Provincial Key Laboratory of Molecular Target & Clinical Pharmacology, School of Pharmaceutical Sciences, Guangzhou Medical University, Guangzhou 511436, China. [3]Guangdong Provincial Key Laboratory of Luminescence from Molecular Aggregates, South China University of Technology, Guangzhou 510640, China. [4]School of Science and Engineering, Shenzhen Institute of Aggregate Science and Technology, The Chinese University of Hong Kong, Shenzhen, Guangdong 518172, China. [5]These authors contributed equally: Wenping Zhu, Ying Li. ✉e-mail: hqpeng@mail.buct.edu.cn; tangbenz@cuhk.edu.cn

PSs have been reported for microbe discrimination, killing, and infection treatments[15]. However, the development of AIE PSs for effective antifungal treatments is still facing tough challenges. First, the molecular design of antifungal AIE PSs generally utilizes elaborate conjugation of electron-donating (D) and -accepting (A) units for enhancing the intersystem crossing (ISC) process[16]. This often requires costly and time-consuming chemical modifications. Second, facile regulation of the antifungal activity of AIE PSs remains unexplored, impeding their further optimization for high-efficiency PDT treatment. As a result, despite a few examples of AIE photosensitizers synthesized for the photodynamic killing of fungi[17], the development of more promising antifungals based on AIE characteristics is currently highly desirable.

Utilizing stereoisomeric engineering to excavate different spatial configurations of one molecule could be an ingenious strategy to develop desirable functional molecules. This is inspired by some clinical drugs, e.g., doxepin and tamoxifen, whose stereoisomeric structures have a pronounced influence on their therapeutic efficacy[18]. On this basis, we envision that using stereoisomers could be a new starting point for exploiting promising AIE PSs. In addition, one of the most attractive features of stereoisomers is their amplified functions in dye aggregates. Therefore, the supramolecular assembly that occurs via noncovalent interactions for the molecular organization is very convenient for engineering AIE stereoisomers. Tetraphenylethene (TPE) and its derivatives as iconic AIE luminogens[19–21] are suitable for stereoisomeric engineering of AIE PSs since bifunctionalized TPE with (Z)- and (E)-configurations can be easily synthesized by McMurry coupling of benzophenone derivatives[22,23]. In our previous work, we reported a pair of TPE stereoisomers and manifested their distinct emissions, morphologies, and material properties in supramolecular assemblies[24,25]. Moreover, we synthesized stereoisomeric TPE amphiphiles, of which the (Z)-amphiphile had more sensitive thermo-responsive behaviors[26]. These findings have demonstrated the feasibility of stereoisomeric engineering for exploiting new functional molecules in material science. However, to the best of our knowledge,

photosensitizers fabricated from pure (Z)- and (E)-configurations of TPE and their self-assemblies have yet to be explored.

In this study, we rationally designed and synthesized new AIE PSs by modifying two cationic 4-vinylpyridine units on TPE (Fig. 1). The pure stereoisomers, namely, (Z)-TPE-EPy and (E)-TPE-EPy were successfully obtained by separating their neutral precursors via column chromatography. Taking advantage of the spatial configurational differences between (Z)- and (E)-TPE-EPy, two photosensitizers with different AIE activities and ROS generation capabilities become available. Furthermore, the (Z)- and (E)-isomers carrying positively charged groups are capable of binding with cucurbit[8]uril (CB[8]) by host-guest interactions with high association constants. The such supramolecular assembly provides chances to engineer our stereoisomeric AIE PSs to realize facile regulation of their PDT activity. In comparison with the monomeric isomers, (Z)-TPE-EPy@CB[8] and (E)-TPE-EPy@CB[8] exhibit enhanced red emission intensity and ROS generation efficiency since the intramolecular motions of the isomers can be restricted in the CB[8] cavity. After systematic experimental and theoretical studies, we evaluated the PDT efficacy of the stereoisomeric AIE PSs and their self-assemblies towards fungi. It turns out that all of them with cationic charges can locate in fungal mitochondria, which is attributed to the negative potential of the mitochondrial membrane in fungus. Gratifyingly, under white light irradiation for 10 min, the killing efficiency of these photosensitizers (5 μM) to *Saccharomyces cerevisiae* and *Candida albicans* can reach almost 100%, of which (Z)- and (E)-TPE-EPy@CB[8] that shielding cationic pyridine by macrocycles show dramatically lowered dark toxicity without sacrificing their significant phototoxicity. Moreover, (Z)- and (E)-TPE-EPy and their self-assemblies can target and eliminate fungi over mammalian cells, which is of great significance for reducing the side effects of PDT in clinical applications. Thus, stereoisomeric engineering assisted by the supramolecular assembly in this study opens up more opportunities for defending against fungal infections and other health crises by introducing new design principles of photosensitizers for PDT.

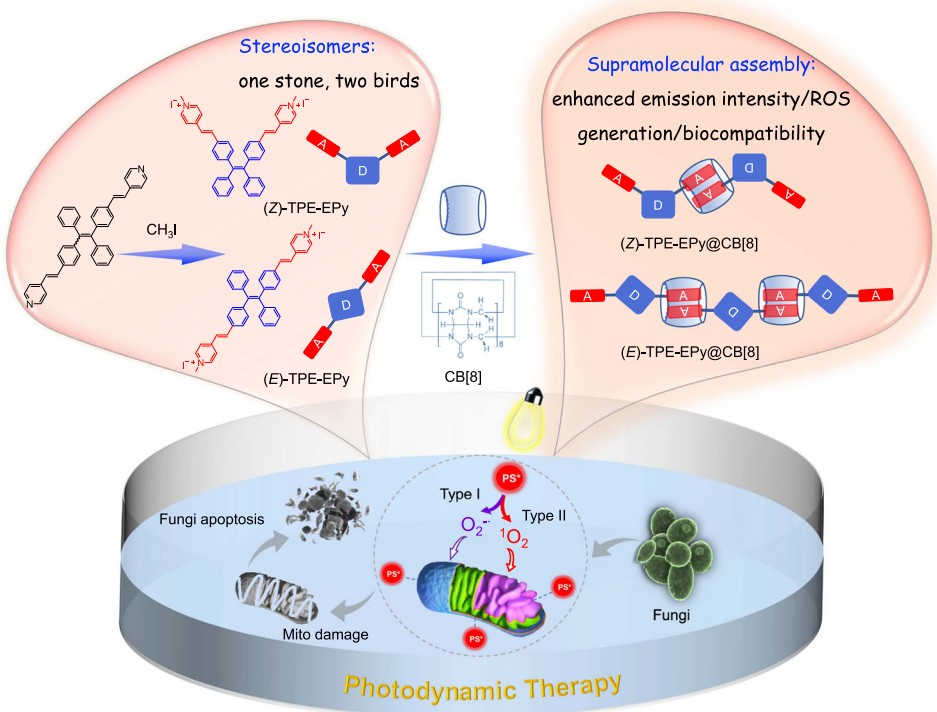

**Fig. 1 | Schematic illustration.** Molecular structures and supramolecular assemblies of (Z)-TPE-EPy and (E)-TPE-EPy for fungal killing.

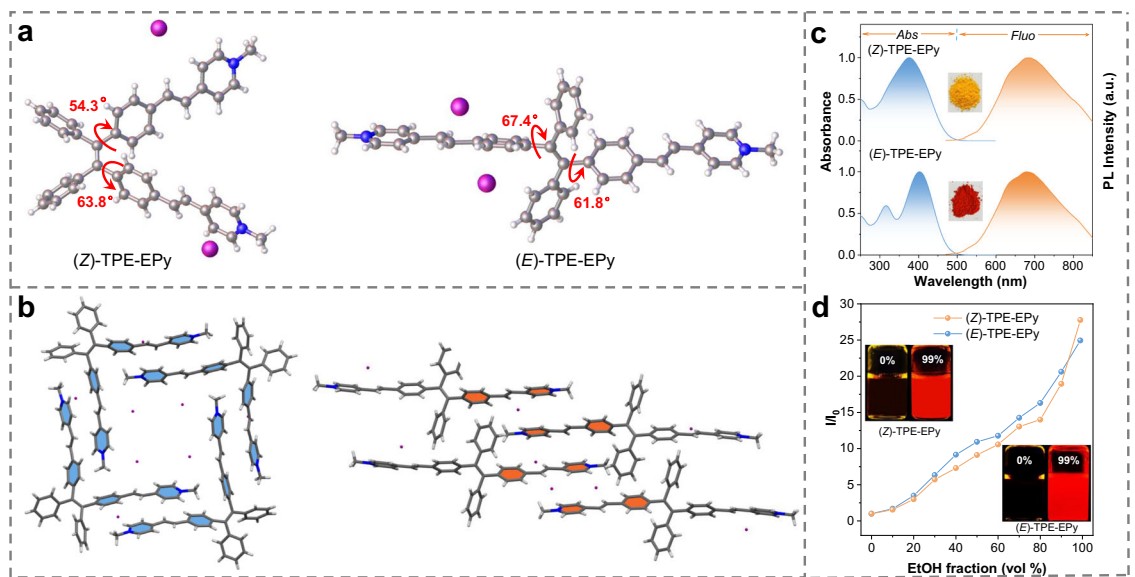

**Fig. 2 | Molecular structures and photophysical properties. a, b** Molecular structures and intermolecular packing of (*Z*)- and (*E*)-TPE-EPy. **c** Normalized absorption and PL spectra of (*Z*)- and (*E*)-TPE-EPy in water, $\lambda_{ex}$ = 450 nm. Inset: Photographs of (*Z*)- (yellow) and (*E*)-TPE-EPy (red) powder. **d** Plot of relative

emission intensity versus the EtOH fraction (vol %) of (*Z*)- and (*E*)-TPE-EPy in H₂O/EtOH mixtures (10 µM, $\lambda_{em}$ = 645 nm). Inset: photographs of (*Z*)- and (*E*)-TPE-EPy in aqueous solution and EtOH/H₂O mixtures (*v:v*, 99:1) taken upon irradiation at 365 nm.

## Results

### Molecular structures and photophysical properties

The synthesis of (*Z*)- and (*E*)-TPE-EPy is depicted in supporting information (Supplementary Figs. 1, 2). The stereoisomers were characterized by ¹H NMR, ¹³C NMR, and high-resolution mass spectroscopies (Supplementary Figs. 3–10). Moreover, single crystals of the (*Z*)- and (*E*)-isomers, obtained by slowly evaporating their dichloromethane/methanol mixed solutions (Supplementary Table 1), unambiguously establish the geometric configurations of the stereoisomers. As shown in Fig. 2a, the torsional angles of styryl pyridine units of the (*Z*)-isomer are 54.3° and 63.8°, respectively. However, the (*E*)-isomer adopts a more twisted D-A configuration with torsion angles of 61.8° and 67.4°. Meanwhile, the (*Z*)- and (*E*)-isomers show different molecular arrangements in crystal states (Fig. 2b). The (*Z*)-isomer stacks head-to-tail to form a regular quadrilateral structure through π–π, CH···π and CH···I⁻ interactions (Supplementary Fig. 11A, 12A)[27]. In contrast, the (*E*)-isomer packs head-to-head to shape a linear structure via CH···π and CH···I⁻ interactions (Supplementary Figs. 11B, 12B). The configurational differences of the stereoisomers lead to their different stacking modes, confirming the rationality of performing stereoisomeric engineering by the supramolecular assembly.

After structural characterizations, the photophysical properties of (*Z*)- and (*E*)-TPE-EPy were studied by investigating their UV-vis absorption spectra and photoluminescence (PL) spectra. The (*Z*)-isomer in water exhibits one main peak at 376 nm, and its (*E*)-cousin has two absorption peaks at 316 and 402 nm (Fig. 2c). Correspondingly, (*Z*)- and (*E*)-TPE-EPy powders in appearance display different colors of yellow and red, respectively. In contrast, emission spectra of the stereoisomers have an almost identical shape with a red fluorescence peaking at 684 nm (Fig. 2c). However, there is a difference between their fluorescence quantum yields (QY) in aqueous solutions (25 nM, QY$_Z$ = 1.96%, QY$_E$ = 2.92%). These discrepant properties of one molecule in changed configurations indicate the feasibility of stereoisomeric engineering for developing diversified functional materials. It should be noted that (*Z*)- and (*E*)-TPE-EPy exhibit good water-solubility due to the ionic characteristic, rationalizing their relatively low QY values in aqueous solutions since TPE derivatives can facilitate non-radiative decay in the solution state. To confirm AIE behaviors of (*Z*)- and (*E*)-TPE-EPy, we increased ethanol fractions, *viz.* the poor solvent in

aqueous solutions of the stereoisomers. The gradually enhanced emission intensities corroborate the occurrence of AIE processes (Fig. 2d, Supplementary Fig. 13).

### Supramolecular assembly

CB[8] is commonly used to encapsulate positively charged guests through host-guest interactions[28–36]. We employed it to self-assemble (*Z*)- and (*E*)-TPE-EPy for regulating the isomers' photophysical properties. ¹H NMR spectra of (*Z*)- and (*E*)-TPE-EPy with CB[8] at different concentrations were investigated to verify their formation of supramolecular assemblies. The specific proton chemical shifts and splitting of the stereoisomers resemble each other (Fig. 3). With the titration of CB[8] from 0 to 4.0 equiv, the H$_a$-H$_f$ proton resonances of (*Z*)-TPE-EPy shift upfield by 0.33–1.07 ppm, indicating their encapsulation in CB[8] cavity with shielding effect. In contrast, H$_g$-H$_i$ on the (*Z*)-isomer shift downfield by 0.09–0.47 ppm, revealing that their locations are outside but near the carbonyl-rimmed portal of CB[8] with a deshielding effect. Similarly, when (*E*)-TPE-EPy is mixed with CB[8] in D₂O (0 to 4.0 equiv), its protons H$_a$-H$_f$ show remarkable upfield shifts from 0.26 to 0.88 ppm. This demonstrates that the 4-styrylpyridine units are wrapped deeply in the CB[8] cavity. The protons H$_g$-H$_i$ downfield shift by 0.17 ppm, which should be attributed to their locations on or near the CB[8] portal. Meanwhile, ¹H NMR spectra of (*Z*)- and (*E*)-TPE-EPy@CB[8] displayed pronounced splitting peaks of CB[8] protons, particularly H$_x$ situated toward the interior of the cavity. This suggests an asymmetric charge density environment for two CB[8] portals, demonstrating the host-guest recognitions of CB[8] with (*Z*)- and (*E*)-TPE-EPy. Additionally, the binding behaviors between the stereoisomers and CB[8] were also estimated by isothermal titration calorimetry (ITC) experiments, which indicated a stoichiometry of 1:1 for (*Z*)- and (*E*)-TPE-EPy@CB[8] assemblies (Supplementary Fig. 14). Based on the above results, we infer that each CB[8] holds two positively charged arms of two neighboring molecules in a head-to-tail stacking pattern to form supramolecular assemblies. Further evidence that corroborates their formation is the hydrodynamic diameter and morphological changes of the stereoisomers after they are assembled with CB[8] (Fig. 4a and Supplementary Figs. 15, 16). The former has a >10-fold increase, and the latter transforms from discrete irregular nanoparticles to well-defined microscopic shapes due to the supramolecular assembly.

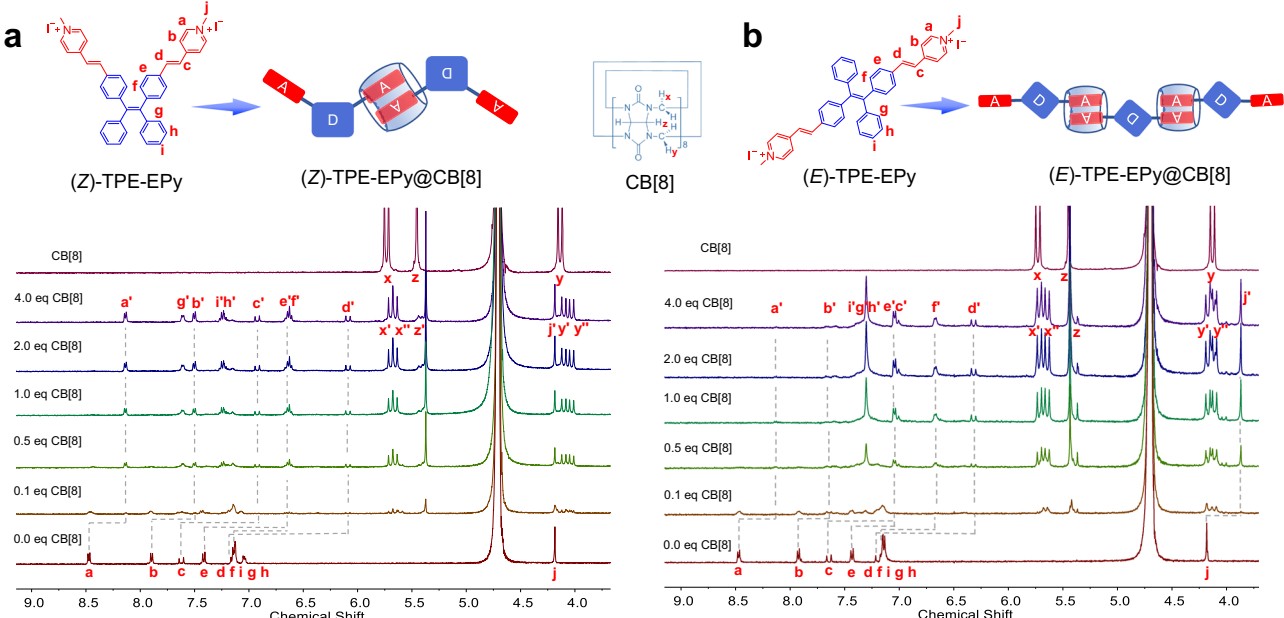

**Fig. 3 | Verification of supramolecular assembly by ¹H NMR titration experiment.** ¹H NMR titration spectra of **a** (*Z*)-TPE-EPy and **b** (*E*)-TPE-EPy with different amounts of CB[8] in D$_2$O. [(*Z*)-TPE-EPy] = [(*E*)-TPE-EPy] = 0.25 mM.

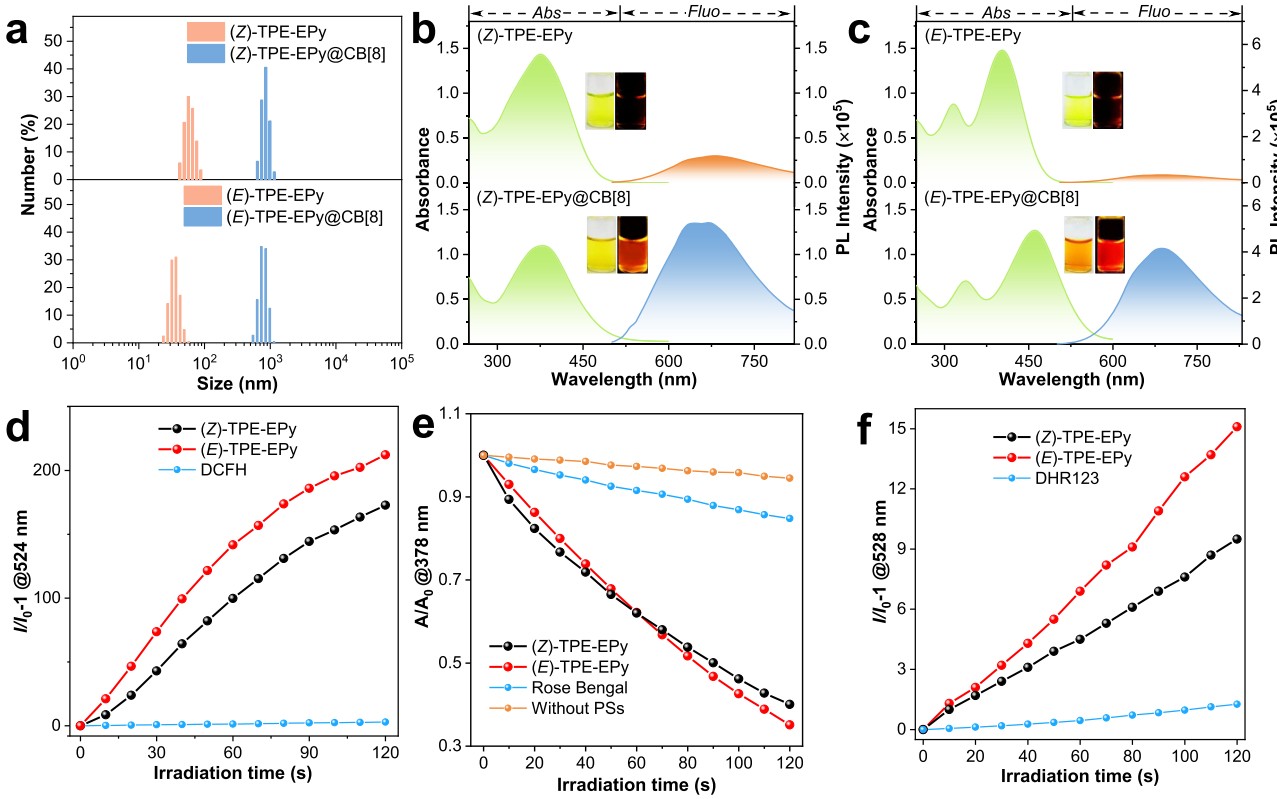

**Fig. 4 | Supramolecular assembly effects and ROS generation of isomers.**
**a** Dynamic light scattering (DLS) data of the stereoisomers and their self-assemblies. The particle dispersion index was 0.189 for (*Z*)-TPE-EPy, 0.141 for (*Z*)-TPE-EPy@CB[8], 0.195 for (*E*)-TPE-EPy, and 0.177 for (*E*)-TPE-EPy@CB[8].
**b** Absorption and PL spectra of (*Z*)-TPE-EPy and (*Z*)-TPE-EPy@CB[8]. **c** Absorption and PL spectra of (*E*)-TPE-EPy and (*E*)-TPE-EPy@CB[8] (λ$_{ex}$ = 450 nm). **d** ROS

generation of (*Z*)- and (*E*)-TPE-EPy (5 μM) upon white light irradiation using DCFH (5 μM) as an indicator. **e** Absorbance decay of ABDA in the absence and presence of different PSs under white light irradiation. [PS] = 5 μM, [ABDA] = 50 μM. **f** Plot of the relative emission intensity (*I*/*I*$_0$) of DHR123 (30 μM) solution containing (*Z*)- or (*E*)-TPE-EPy (5 μM) versus the irradiation time. The white light irradiation power is 26 mW cm$^{-2}$.

We further investigated UV–vis absorption and PL of (Z)- and (E)-TPE-EPy@CB[8]. (Z)-TPE-EPy exhibits a broad absorption peaked at 376 nm. With the gradual addition of CB[8] from 0 to 4 equiv, the absorption peak was decreased and narrowed, which could be due to the restriction of rotation of the styrylpyridine units on (Z)-TPE-EPy@CB[8] (Fig. 4b, Supplementary Fig. 17A). Meanwhile, the corresponding fluorescence intensity enhances distinctly (Supplementary Fig. 18A). In addition, the QY$_Z$ with CB[8] can increase from 1.96 to 3.44% even at a very low concentration of 25 nM. These results are consistent with the AIE characteristics of the stereoisomers. Detailly, the intramolecular motions of (Z)-TPE-EPy are restricted by host-guest interactions, suppressing the nonradiative energy dissipation and leading to stronger fluorescence signals. By comparison, with the CB[8] (from 0 to 4 equiv) gradually adding into the (E)-TPE-EPy aqueous solution, the absorption of (E)-TPE-EPy decreases and also red-shift with the two absorbance maxima shifting from 316 and 402 nm to 338 and 462 nm, accompanied by a visible solution color change from bright green to yellow (Fig. 4c and Supplementary Fig. 17B). This can be explained by the host- enhanced π–π interaction, i.e. extended π-conjugation of (E)-TPE-EPy when its two opposite ends on the linear structure are incorporated into the CB[8] cavity. This is why CB[7] with a smaller cavity that encapsulates only one (E)-isomer end group can exclude the pronounced red-shift absorption (Supplementary Fig. 19B). However, the corresponding PL maximum shift of (E)-TPE-EPy@CB[8] in water is neglectable, which may result from a compromise between the weakened polarity microenvironment for the isomer with a D-A structure and its extended π-conjugation in CB[8]. (E)-TPE-EPy@CB[8] also exhibits sharply enhanced fluorescence intensity, with the QY increased from 2.9 to 6.3% due to the AIE process (Fig. 4c and Supplementary Fig 18B).

It is clear that (E)-TPE-EPy displays a higher PL intensity enhancement than its (Z)-cousin, indicating a severer restriction of CB[8] on the (E)-configurations. By fluorescence titration experiments, the binding constants ($K_a$) of (Z)- and (E)-TPE-EPy with CB[8] were determined to be $5.8 \times 10^4$ and $3.6 \times 10^5$ L mol$^{-1}$, respectively, which also rationalized the severer restriction of CB[8] on the (E)-isomer (Supplementary Fig. 18).

### ROS generation of the stereoisomeric AIE PSs

The AIE-active (Z)- and (E)-TPE-EPy have positively charged 4-vinylpyridine groups for electron-accepting in the D-A structure, which is helpful in lowering the energy gap between the excited singlet state (S₁) and the triplet state (T₁), thereby can realize efficient ROS

generation by enhancing the ISC process. To verify this, a commonly used ROS indicator dichlorofluorescein (DCFH) was employed to investigate the ROS generation capabilities of (Z)- and (E)-TPE-EPy. Upon white light irradiation (26 mW cm$^{-2}$), DCFH alone is nearly non-fluorescent (Fig. 4d). However, its fluorescence intensity increases rapidly in the presence of the (Z)- and (E)-isomer (5 μM), demonstrating that DCFH is oxidized by ROS that produced from the stereoisomers. In comparison, (E)-TPE-EPy has higher ROS generation efficiency than the (Z)-configuration. It is well-known that PSs can produce different types of ROS, including superoxide radical $O_2^{\cdot-}$, hydroxyl radical OH·, highly reactive $^1O_2$, or others depending on two distinct mechanisms called Type-I and Type-II processes[37–41]. To identify the ROS types, a $^1O_2$ probe 9,10-anthracenediyl-bis(methylene) dimalonic acid (ABDA) was used to evaluate the $^1O_2$ generation capabilities of the stereoisomeric PSs. Under white light irradiation, the absorbance of ABDA decreases sharply with (Z)- or (E)-TPE-EPy in the aqueous solution (Fig. 4e and Supplementary Fig. 20). This is due to the decomposition of ABDA during the generation of $^1O_2$ via the Type-II process of the isomers. The consumption rate of ABDA was calculated to be 30.0 and 32.5 nmol min$^{-1}$ for (Z)- and (E)-TPE-EPy, respectively, higher than that of the commercial photosensitizer Rose Bengal (7.6 nmol min$^{-1}$), indicating the excellent performance of the stereoisomers for highly reactive $^1O_2$ generation. It is worth noting that $^1O_2$ generation capabilities of the two isomers are inconsistent with the results obtained from the DCFH indicator, suggesting that the PSs also produce other types of ROS through the Type-I process. Therefore, we further detected the primary product of the Type-I process, i.e. $O_2^{\cdot-}$, by using dihydrorhodamine 123 (DHR123) as the indicator. Under white light irradiation, remarkable fluorescence intensity enhancement of DHR123 was observed in the presence of the stereoisomers (Fig. 4f). Obviously, the higher ROS generation efficiency of (E)-TPE-EPy than that of (Z)-TPE-EPy originates from its better $O_2^{\cdot-}$ generation capability. To further confirm the generation of $^1O_2$ and $O_2^{\cdot-}$ from (Z)- and (E)-TPE-EPy, we additionally study them using electron spin resonance (ESR) spectroscopy. 2,2,6,6-tetramethylpiperidine (TEMP) serves as a spin trapper for $^1O_2$ identification. Upon light-irradiating the solution of the stereoisomers and TEMP, a typical paramagnetic absorption is observed and matches with the $^1O_2$ signal (Fig. 5a, b). Thereafter, 5,5-dimethyl-1-pyrroline-Noxide (DMPO) is employed as a spin-trap agent, which successfully verifies the generation of $O_2^{\cdot-}$ species (Fig. 5c, d). According to the above results, we can conclude that (Z)- and (E)-TPE-EPy are capable of producing singlet oxygen and superoxide free radicals, rendering the stereoisomers promising AIE PSs for image-

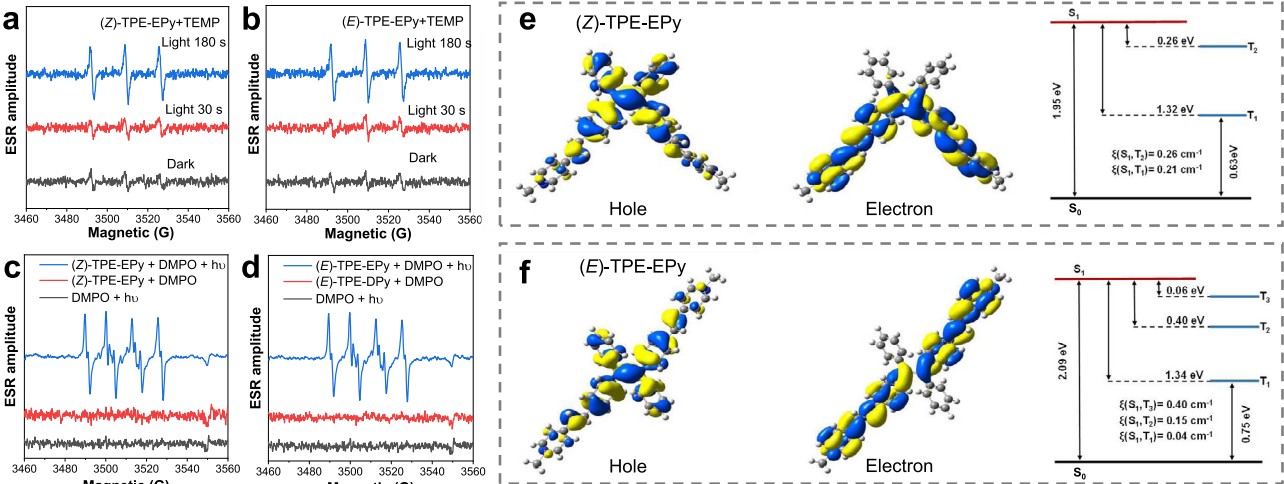

**Fig. 5 | ESR spectra and theoretical calculation of isomers ROS generation.** ESR spectra to detect $^1O_2$ generation from **a** (Z)- and **b** (E)-TPE-EPy by irradiating them at different periods and using TEMP as the spin trapper. ESR spectra to detect $O_2^{\cdot-}$ from **c** (Z)- and **d** (E)-TPE-EPy under irradiation, using DMPO as the spin-trap agent. The NTOs and calculated energy level diagram between singlet and triplet states of **e** (Z)- and **f** (E)-TPE-EPy.

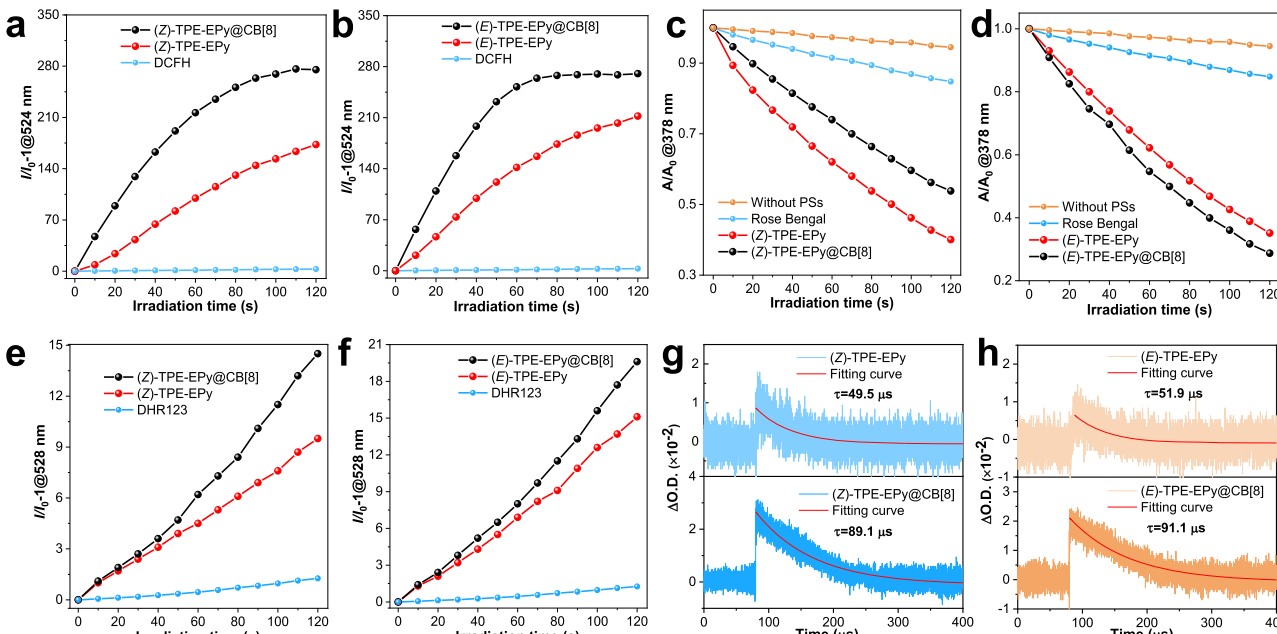

**Fig. 6 | Effect of supramolecular assembly on ROS.** ROS generation from **a** (*Z*)-TPE-EPy, **b** (*E*)-TPE-EPy, and their self-assemblies ([Stereoisomer] = 5 μM, CB[8] = 4 equiv) upon white light irradiation, using DCFH (5 μM) as an indicator. **c, d** Absorbance decay of ABDA (50 μM) in the absence and presence of different PSs under white light irradiation. **e, f** plot of the relative emission intensity (I/I₀) of DHR123 solution (30 μM) containing different PSs versus the irradiation time under white light irradiation (26 mW cm⁻²). Laser flash photolysis to measure the lifetime of the triplet states of **g** (*Z*)-TPE-EPy and (*Z*)-TPE-EPy@CB[8], **h** (*E*)-TPE-EPy and (*E*)-TPE-EPy@CB[8] in ultrapure water.

guided PDT. To elucidate the superior ROS generation capability of the (*E*)-isomer to that of (*Z*)-TPE-EPy, theoretical calculations were performed by using Gaussian09 based on time-dependent density functional theory (TD-DFT). We first calculated the natural transition orbitals (NTOs) of the stereoisomers, which display similar results that the hole NTOs are mainly distributed on the TPE core. In contrast, the electron NTOs are localized at the 4-vinylpyridine arms (Fig. 5e, f). This highly separated charge distribution on the stereoisomers leads to narrow energy gaps between the singlet and triplet states, implying efficient ISC processes for ROS generation from oxygen molecules. We thus calculated the energy levels of singlet and triplet states of (*Z*)- and (*E*)-TPE-EPy. It was found that the (*E*)-isomer possesses an additional ISC channel through the interaction between $S_1$ and the high-lying triplet state $T_3$, showing an energy gap as low as 0.06 eV. At the same time, spin-orbit coupling (SOC) is also responsible for the enhancement of ISC. The SOC value between $S_1$ and $T_3$ for the (*E*)-TPE-EPy was calculated to be 0.40 cm⁻¹, which is larger than that of other ISC channels. Therefore, the efficient ISC process from $S_1$ to $T_3$ should contribute to the superior ROS generation capability of the (*E*)-isomer.

Supramolecular assembly successfully enhanced the optical properties of the stereoisomers. We further revealed the effect of supramolecular assembly on ROS generation of the (*Z*)- and (*E*)-configurations. As shown in Fig. 6a, b, Supplementary Fig. 21, the stereoisomers exhibit improved ROS generation efficiency after they are self-assembled with the CB[8] host, which is consistent with the results mentioned above of fluorescence enhancement. This is rationalized by the excited state energy flow from nonradiative decay to the fluorescence pathway and ISC process due to the supramolecular assembly. Under white light irradiation, the absorbance of ABDA decreases sharply in the presence of the stereoisomers with 4 equiv of CB[8] (Fig. 6c, d), corresponding to consumption rates of 23.0 and 35.6 nmol min⁻¹ for (*Z*)- and (*E*)-TPE-EPy@CB[8], respectively. To our surprise, the consumption rate of (*Z*)-TPE-EPy@CB[8] to ABDA is lower than that of the monomeric (*Z*)-TPE-EPy (30 nmol min⁻¹), which is contradictory to

the result obtained by using DCFH as the indicator. Thus, we assume that (*Z*)-TPE-EPy@CB[8] can generate free radicals more effectively through the type-I process. By employing DHR123, it is demonstrated that the $O_2^{-\cdot}$ generation efficiency of (*Z*)-TPE-EPy@CB[8] is increased by 58%, which is higher than that of its (*E*)-configuration (30%) with host-guest interactions (Fig. 6e, f). In other words, the binding with CB[8] results in suppressed $^1O_2$ but enhanced $O_2^{-\cdot}$ generation efficiency of the (*Z*)-isomer. Both the capabilities of (*E*)-TPE-EPy to produce $^1O_2$ and $O_2^{-\cdot}$ are promoted by the supramolecular assembly. These results suggest that supramolecular assembly, as a simple but efficient strategy, is able to regulate the ROS generation capabilities of the photosensitizers for optimized PDT performance.

To clarify the underlying mechanisms for the overall improved ROS generation from (*Z*)- and (*E*)-TPE-EPy@CB[8], laser flash photolysis was employed to measure the lifetime of the triplet states of the stereoisomers and their supramolecular assemblies. The lifetimes of the triplet states of (*Z*)- and (*E*)-TPE-EPy were measured to be 49.5 and 51.9 μs, respectively (Fig. 6g, h). After binding with CB[8], the stereoisomers exhibit a prolonged lifetime of their triplet states with values measured to be 89.1 and 91.1 μs (Fig. 6g, h). The prolonged lifetime of triplet states benefits energy transfer or electron transfer from PSs to other chemicals, thereby improving ROS generation efficiency. This explains the overall improved ROS generation efficiency of (*Z*)- and (*E*)-TPE-EPy@CB[8].

## Mitochondria-specific PDT killing of fungi

The negative surface and mitochondrial membrane potential of fungus imply that AIE PSs bearing inherent pyridine cations may show good specificity to the fungal mitochondria via electrostatic attractions. To confirm this, the co-staining experiment of the PSs and a commercial mitochondrial probe Mito-Tracker Deep Red (DR), was performed for imaging of *Saccharomyces cerevisiae* (*S. cerevisiae*) and *Candida albicans* (*C. albicans*). As shown in Fig. 7, the colocalization images of these two fungi show that the red fluorescence from the stereoisomeric PSs and their supramolecular assemblies completely overlaps the emission

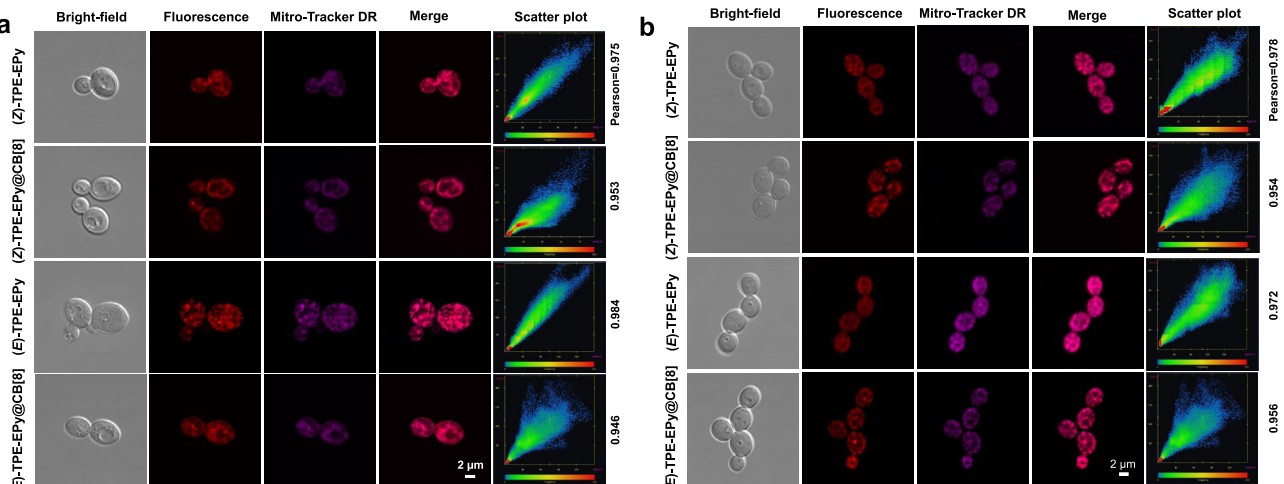

**Fig. 7 | Confocal microscopy images.** Confocal microscopy imaging of **a** *S. cerevisiae* and **b** *C. albicans* labeled with the AIE PSs ([Stereoisomer] = 2 μM) and their colocalization with Mito-Tracker DR (1 μM). Experiments in **a** and **b** were performed three times independently, representative images are shown.

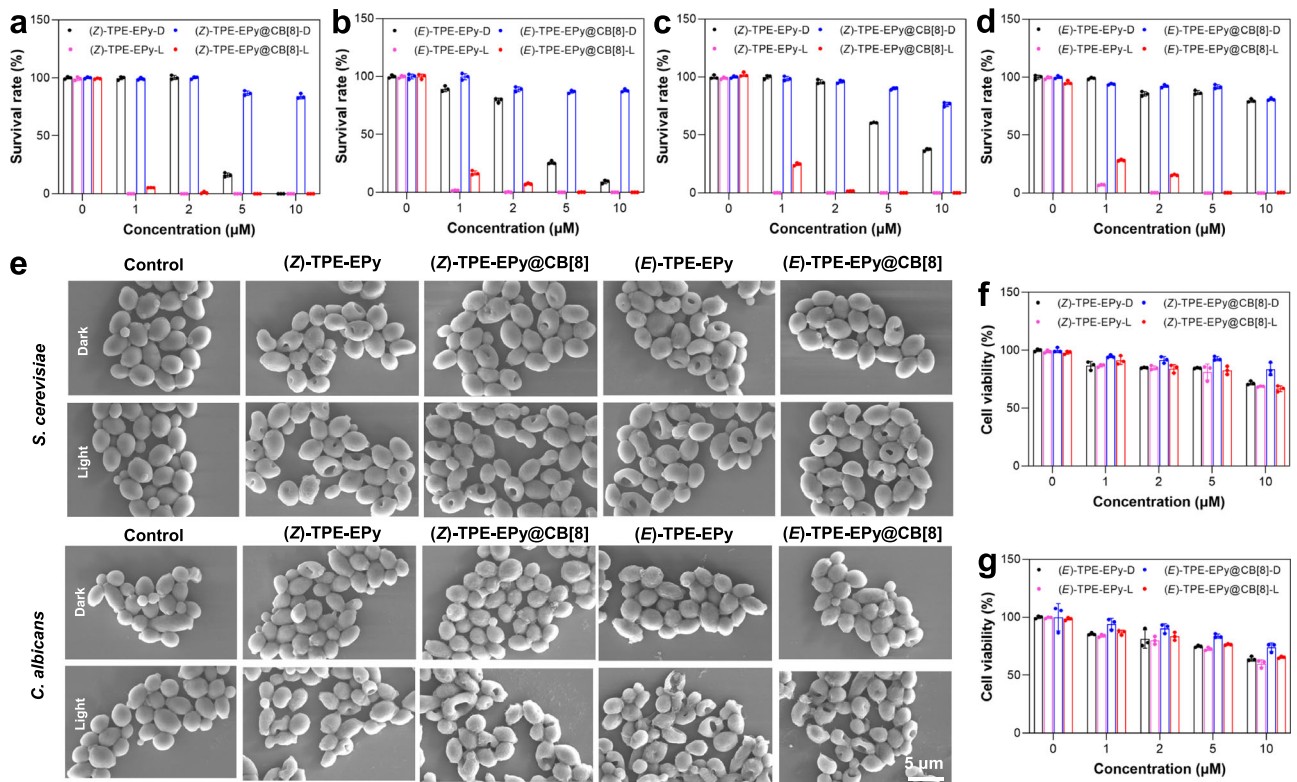

**Fig. 8 | Antifungal performance of the AIE PSs.** Survival rate of **a**, **b** *S. cerevisiae* and **c**, **d** *C. albicans* in darkness or upon light irradiation with treatments of the AIE PSs at different concentrations. **e** SEM images of *S. cerevisiae* and *C. albicans* incubated with the stereoisomers (10 μM) in darkness or upon light irradiation. Experiments were performed three times independently, representative images are shown. **f** Cell viability of HUVECs after incubation with (*Z*)-configurational PSs and **g** (*E*)-configurational PSs at different concentrations in darkness or upon white-light illumination. Data in **a**–**d**, **f**, **g** are presented as mean ± SD (*n* = 3 biological independent samples).

from Mito-Tracker DR. The corresponding high Pearson's correlation coefficients (0.94–0.98) between the PSs and Mito-Tracker DR also fully confirm the high specificity of (*Z*)-TPE-EPy, (*E*)-TPE-EPy, (*Z*)-TPE-EPy@CB[8], and (*E*)-TPE-EPy@CB[8] to the mitochondria of fungi. We evaluated the photodynamic therapeutic effect of these PSs on *S. cerevisiae* and *C. albicans* by using a traditional plate counting method (Supplementary Figs. 22–25). The survival rate of *S. cerevisiae* with AIE PSs (1–10 μM) decreases in darkness (Fig. 8a, b). More than 70% of *S.*

*cerevisiae* are killed at a concentration of 5 μM (*Z*)- or (*E*)-TPE-EPy, and as the stereoisomers reached 10 μM, almost no fungal colony is observed on the agar (Supplementary Fig. 26). This result indicates that high concentrations of the stereoisomers have severe dark toxicity to *S. cerevisiae*. This is not surprising since the diploid cationic pyridine of (*Z*)- and (*E*)-TPE-EPy can efficiently damage cell membranes and depolarize the mitochondrial membrane potential, thereby exerting potent cytotoxicity.

Under the same conditions, the survival rate of *C. albicans* is higher than that of *S. cerevisiae*, suggesting that *C. albicans* is more tolerant to the stereoisomeric PSs in darkness (Fig. 8c, d). On the other hand, under white light irradiation for 10 min and at a very low concentration of the stereoisomers (1.0 μM), the killing efficiencies of (*Z*)- and (*E*)-TPE-EPy to *S. cerevisiae* can reach about 99.9 and 98.3%, respectively, and is about 99.8 and 92.8% for *C. albicans*. The antifungal effect of (*Z*)-TPE-EPy is slightly better than that of (*E*)-TPE-EPy, which may be due to its higher Zeta potential that results in higher binding efficiency with fungi (Supplementary Fig. 27). These results suggest that (*Z*)- and (*E*)-TPE-EPy possess satisfactory phototoxicity because of their high ROS generation capabilities. However, an ideal photosensitizer should exhibit satisfying phototoxicity for high PDT efficiency but negligible dark toxicity for treatment safety. Considering that CB[8] is able to encapsulate 4- styrylpyridine units and thus shield the cationic pyridine, we hypothesize that (*Z*)- and (*E*)-TPE-EPy@CB[8] can be employed to reduce dark toxicity of the stereoisomers. Meanwhile, Pearson's correlation coefficients demonstrate that the shielding effect has no significant effect on the mitochondrial targeting capabilities of the supramolecular PSs, presumably due to their higher Zeta potentials resulting from self-assembly of (*Z*)- and (*E*)-isomers (Supplementary Fig. 27). To test this hypothesis, these supramolecular PSs were incubated with *S. cerevisiae* and *C. albicans* in darkness (Fig. 8a–d). The improved survival rate of the fungus indicates that after being encapsulated by CB[8], the stereoisomeric PSs display obviously lowered dark toxicity. For example, the survival rate of *S. cerevisiae* with (*E*)-TPE-EPy (10 μM) in the absence and presence of CB[8] is 9.1 and 88.0%, respectively. Meanwhile, *S. cerevisiae* and *C. albicans* can be killed effectively if the groups are irradiated by white light, even when the PS concentration is as low as 2 μM, indicating satisfying phototoxicity of the supramolecular PSs.

To further evaluate the PDT performance of the PSs, we utilized scanning electron microscopy (SEM) to gain insights into the morphological changes of the fungus. In the absence of the PSs, the morphologies of *S. cerevisiae* remain intact with well-defined borders and smooth bodies both in darkness and under light irradiation (Fig. 8e). However, the cell membranes of *S. cerevisiae* are damaged with holes after treatment with (*Z*)- or (*E*)-TPE-EPy (10 μM) in darkness, confirming the dark toxicity of the stereoisomeric PSs. Under the same conditions, the morphology of *C. albicans* can remain intact due to its tolerance to (*Z*)- and (*E*)-TPE-EPy. Once the two kinds of fungus with stereoisomeric PSs are exposed to white light, their cell walls can be vigorously damaged, collapsed, and fused, suggesting that *S. cerevisiae* and *C. albicans* are killed by ROS generated from (*Z*)- or (*E*)-TPE-EPy. In contrast to the monomeric PSs, when *S. cerevisiae* and *C. albicans* are treated with (*Z*)- and (*E*)-TPE-EPy@CB[8], their morphologies are intact owing to the inhibited dark toxicity. Furthermore, the satisfying phototoxicity of (*Z*)- and (*E*)-TPE-EPy@CB[8] is demonstrated by the dramatically changed morphologies of *S. cerevisiae* or *C. albicans* upon white light irradiation. These direct observations from SEM are consistent with the antifungal results, revealing that stereoisomeric engineering of PSs by supramolecular assembly can be a promising approach to developing ideal photosensitizers with good biocompatibility and high antifungal activities. It is worth mentioning that our PSs also have good binding efficiency to bacteria and can exhibit surpass 98 and 99% phototoxicity to *Staphylococcus aureus* (*S. aureus*) and *Escherichia coli* (*E. coli*) at the concentration of 0.5 μM and 5 μM, respectively (Supplementary Figs. 28–33).

*S. cerevisiae* and *C. albicans* can be clearly visualized with bright red fluorescence after being incubated with AIE PSs (Fig. 7). However, under the same staining conditions, almost no fluorescence signals were detected in human umbilical vein endothelial cells (HUVECs) (Supplementary Fig. 34), implying that the stereoisomeric PSs and their assemblies rarely enter normal mammalian cells. To further investigate the cytotoxicity of the stereoisomeric PSs and their

supramolecular assemblies, the live/dead cell co-staining assays were performed by using Calcein-AM with green fluorescence and propidium iodide (PI) with red fluorescence for live cells and dead cells, respectively. Being similar to the control group, the HUVECs treated with the stereoisomeric PSs or their assemblies exhibit obvious green fluorescence, but show no occurrence of red fluorescence under dark or light irradiation, indicating the high survival rate of mammalian cells (Supplementary Figs. 35–38). The cytotoxicity was further quantitatively evaluated by the standard method of the 3-(4,5-dimethylthiazol-2-yl)−2,5-diphenyltetrazolium bromide (MTT) assay (Fig. 8f, g). It is demonstrated that the stereoisomeric PSs and their supramolecular assemblies have low cytotoxicity at the effective concentration for killing fungus. In addition, hemolysis analysis in red blood cells demonstrates that both the stereoisomers and their supramolecular assemblies have good blood compatibility (Supplementary Fig. 39). Such good biocompatibility to HUVECs and red blood cells implies that our AIE PSs are ideal photosensitizers for achieving selective killing of fungus with low side effects in clinical applications.

## Discussion

In summary, we have developed promising photosensitizers by supramolecular assembly-assisted stereoisomeric engineering. (*Z*)- and (*E*)-TPE-EPy are successfully obtained by modifying two 4-vinylpyridine groups on TPE. The cationization of the stereoisomers not only enables separated charge distribution for efficient ROS generation but also realizes targeting of fungal mitochondria for antifungal PDT killing. It is demonstrated that both (*Z*)- and (*E*)-TPE-EPy possess AIE characteristics and $^1O_2/O_2^{-•}$ generation capabilities, among which the (*E*)-configuration is better due to its additional ISC channel from $S_1{\to}T_3$. The host-guest interactions between the stereoisomers and CB[8] yield (*Z*)- and (*E*)-TPE-EPy@CB[8] with optimized photophysical properties and improved ROS generation efficiency owing to the restriction of intramolecular motions of the stereoisomers in the cavity of CB[8]. In the photodynamic antifungal experiments, by binding with CB[8], the dark toxicity caused by the diploid cationic pyridine on the stereoisomeric PSs is dramatically inhibited without sacrificing the phototoxicity for antifungal PDT treatment. This study is a demonstration of stereoisomeric engineering of AIE PSs based on (*Z*)- and (*E*)-configurations. In addition, to simplify the synthesis by using different configurations of one molecule, we also confirm the potential of supramolecular assembly as a convenient approach to optimize PDT effects. We anticipate that this line of research will open up more opportunities to manufacture new medical materials and provide a new scheme to decrease the defects of some clinical antifungal drugs.

## Methods

### Fluorescence titration experiments

The binding constant ($K_a$) was calculated according to the Scatchard equation: $K_a$: binding constant; n: the number of dye sites per phosphate; r: the ratio of the concentration of the binding AIE -PSs to the concentration of CB[8]; $C_f$: concentration of free AIE PSs. The concentration of the binding compound ($C_b$) was calculated by the following formula:

$$\frac{r}{c_f} = k_a \times n - k_a \times r \quad (1)$$

$C_t$: the total concentration of (*Z*)-TPE-EPy or (*E*)-TPE-EPy; F: the observed fluorescence intensity at a given CB[8] concentration; $F_0$: the fluorescence intensity of the AIE -PSs only (no CB[8]); $F_{max}$: the fluorescence intensity of the total binding compound. And the $C_f$ was the difference value of $C_t$ to $C_b$[42].

$$c_b = c_t \times \frac{F - F_0}{F_{max} - F_0} \quad (2)$$

## DCFH assay

Compound 2′,7′-dichlorodihydrofluorescein (DCFH) was employed as an indicator for the detection of ROS. The concentration of DCFH was 40 μM. After being saturated with oxygen, AIE PSs were added into the solution of DCFH for ROS generation measurement. The generation of ROS was detected by monitoring the emission at 522 nm every 10 s. The excitation wavelength was 488 nm.

## ABDA Assay

Compound 9,10-Anthracenediyl-bis(methylene)dimalonic acid (ABDA) was used to detect singlet oxygen. After being saturated with oxygen, AIE PSs were added into the solution of ABDA (50 μM, PBS), followed by irradiation under light. Singlet oxygen generation was detected by monitoring the absorbance at 378 nm every 10 s.

## DHR123 assay

Compound methyl 2-(3,6-diamino-9H-xanthen-9-yl) benzoate (DHR123) was used to detect superoxide radicals. After being saturated with oxygen, PS was added into the solution of DHR123 (30 μM), followed by irradiation under light. The generation of superoxide radicals was detected by monitoring the emission at 525 nm every 10 s. The excitation wavelength was 488 nm.

## Theoretical calculations

All the $S_0$ and $T_1$ geometries of (Z)- and (E)-TPE-EPy are optimized using the density functional theory (DFT) methodology, implemented with the wB97X-D range separation functional and 6−31G(d) basis set, and further single point energies are calculated using 6−311G(d,p) basis set. Meanwhile, the vertical excitation energies (VEEs) and spin-orbit coupling (SOC) values of these geometries are calculated using the time-dependent density functional theory (TD-DFT) at the TD-wB97X-D/6-311G(d,p) level of theory with the closed shell reference state. All the DFT and TD-DFT calculations are performed using Gaussian 09 package, Revision D.01, and the calculation of SOC values was implemented by using the code named PySOC.

## Bacterial/fungal strains and growth conditions

*Staphylococcus aureus* (S. aureus) ATCC 29213 strain and *Escherichia coli* (*E. coli*) ATCC 25922 strain used herein were purchased from American type culture collection (ATCC). *Saccharomyces cerevisiae* (*S. cerevisiae*) P11 strain and *Candida albicans* (*C. albicans*) ATCC 10231 strain used herein were purchased from China General Microbiological Culture Collection Center. *S. aureus* and *E. coli* were routinely grown on Luria-Bertani (LB) broth (Sangon Biotech, China) medium or LB broth agar plate at 37 °C. *S. cerevisiae* and *C. albicans* were grown on Yeast Extract Peptone Dextrose (YPD) medium (Sangon Biotech, China) or YPD agar plate at 37 °C.

## Zeta potential measurements

*S. cerevisiae* and *C. albicans* were incubated with AIE PSs at 37 °C for 30 min, respectively. After centrifugation at 4722 relative centrifugal force (×*g*) for 3 min, the fungi were collected and dispersed in PBS for zeta potential measurements. As for the blank control group, fungi without PSs were treated under the same conditions.

## Fungal culturing and imaging

*S. cerevisiae* and *C. albicans* were cultured in the yeast medium at 37 °C with a shaking speed of 200 rpm. Fungi were harvested by centrifuging at 4722 relative centrifugal force (×*g*) for 2 min. *S. cerevisiae* and *C. albicans* ($OD_{600}$ = 1) were added with 1 mL phosphate-buffered saline (PBS) containing 2 μM AIE PSs and 1 μM Mito-tracker Deep Red (DR). The mixed solutions were incubated at 37 °C with a shaking speed of 200 rpm for 10 min. 1 μL of stained fungi solution was transferred to a piece of glass slide and then covered by a coverslip, the images were collected using a CLSM (LSM900, Carl Zeiss, Germany). $\lambda_{ex}$ = 405 nm,

$\lambda_{em}$ = 600−640 nm for (Z)-TPE-EPy, (E)-TPE-EPy, and (Z)-TPE-EPy@CB[8]; $\lambda_{ex}$ = 488 nm, $\lambda_{em}$ = 600−640 nm for (E)-TPE-EPy@CB[8]; $\lambda_{ex}$ = 643 nm, $\lambda_{em}$ = 650−700 nm for Mito-Tracker DR.

## Antifungal assay

*S. cerevisiae* and *C. albicans* ($OD_{600}$ = 0.1) were dispersed in 1 mL of different concentrations (0, 1, 2, 5, 10 μM) of AIE PSs PBS solution and then incubated at 37 °C with a shaking speed of 200 rpm for 30 min. The AIE PSs/fungi mixed solution in the darkness or under white light irradiation (16 mW cm$^{-2}$) for 10 min, respectively. 100 μL diluted fungi was sprayed on the yeast agar plate, followed by culturing at 37 °C for 24 h before colony forming units (CFU) counting and taking photos. Triplicate analyses of each sample were performed, and each experiment was carried out in duplicate.

## HR-SEM analysis

Fungi cultured in yeast medium were collected and resuspended into PBS after 12 h. After treatment of fungi ($OD_{600}$ = 1) with AIE PSs (10 μM) at 37 °C with a shaking speed of 200 rpm for 30 min. The AIE PSs/fungi mixed solution in the darkness or under white light irradiation (16 mW cm$^{-2}$) for 10 min, respectively. The fungi were collected and fixed with 2.5% glutaraldehyde solution overnight. The glutaraldehyde was removed by centrifugation, and the fungi were washed with PBS for 2 times. Then the fungi were dehydrated by 30, 50, 70, 80, 90, 95, and 100% (v/v in water) ethanol in sequence for 15 min each. 2 μL of fungi suspensions was added onto clean silicon slices followed by naturally drying in the air. The specimens were coated with Au before the analysis of HR-SEM (APREOS, Thermo scientific, Netherlands).

## Bacteria culturing and imaging

*S. aureus* and *E. coli* were cultured in the Luria-Bertani (LB) broth medium at 37 °C with a shaking speed of 200 rpm. Bacteria were harvested by centrifuging at 4722 relative centrifugal force (×*g*) for 2 min. *S. aureus* and *E. coli* (OD600 = 0.5) were added with 1 mL phosphate-buffered saline (PBS) containing 5 μM AIE PSs and incubated at 37 °C with a shaking speed of 200 rpm for 10 min. 1 μL of stained bacteria solution was transferred to a piece of glass slide and then covered by a coverslip. The images were collected using a CLSM (LSM900, Carl Zeiss, Germany). $\lambda_{ex}$ = 405 nm, $\lambda_{em}$ = 600−700 nm for (Z)-TPE-EPy, (E)-TPE-EPy, and (Z)-TPE-EPy@CB[8]; $\lambda_{ex}$ = 488 nm, $\lambda_{em}$ = 600−700 nm for (E)-TPEEPy@CB[8].

## Antibacteria assay

*S. aureus* and *E. coli* ($10^8$ CFU mL$^{-1}$) were dispersed in 1 mL of different concentrations (*S. aureus*: 0, 0.25, 0.5, 1 μM; *E. coli*: 0, 1, 2, 5 μM) of AIE PSs PBS solution and then incubated at 37 °C with a shaking speed of 200 rpm for 30 min. The AIE PSs/bacteria mixed solution in the darkness or under white light irradiation (16 mW cm$^{-2}$) for 10 min, respectively. 100 μL diluted ($10^4$ CFU mL$^{-1}$) bacteria was sprayed on the LB agar plate, followed by culturing at 37 °C for 24 h CFU counting and taking photos. Triplicate analyses of each sample were performed, and each experiment was carried out in duplicate.

## Hemolysis assay

The mouse blood samples were centrifuged at 1500 rpm for 15 min and washed with saline for three times to get erythrocytes. The obtained erythrocytes were re-diluted into 4% suspension. The suspensions mentioned above were mixed with 10 μM (Z)-TPE-EPy, (Z)-TPE-EPy@CB[8], (E)-TPE-EPy, (E)-TPEEPy@CB[8] and incubated at 37 °C for 3 h. Water was used as a positive control in this experiment. The supernatants were acquired through centrifugation at 12,000 rpm for 15 min and then measured absorbance at 540 nm by UV−vis spectrophotometer. The animal experiments were approved by the Institutional Animal Care and Use Committee of Guangzhou Medical

University and performed in compliance with NIH guidelines for the care and use of laboratory animals (protocol # GY2020-120).

## Cell culturing and imaging

HUVECs used herein were purchased from ATCC cell banks (ATCC® CRL-1730.). HUVECs were cultured in Dulbecco's minimum essential medium (DMEM) (Gibco, USA) containing 10% fetal bovine serum (FBS) (Gibco, USA) and 1% penicillin (100 units mL$^{-1}$) and streptomycin (100 µg mL$^{-1}$) in a 5% $CO_2$ humidity incubator at 37 °C. The cells were subcultured every three days. HUVECs ($1 \times 10^6$ cells mL$^{-1}$) were seeded in a confocal dish and cultured for 24 h, cells were added with 2 and 10 µM for 10 min in a 5% $CO_2$ humidity incubator at 37 °C.Then, cells were washed with PBS buffer and imaged using a CLSM (LSM900, Carl Zeiss, Germany).

## Cell viability assay

HUVECs were grown on a 96-well plate at a density of $1 \times 10^4$ cells per well (200 µL) and incubated for 24 h. After incubation with AIE PSs at different concentrations of 0, 1, 2, 5, and 10 µM for 30 min, the cells were exposed to white light irradiation (16 mW cm$^{-2}$) for 10 min or in the darkness. They were then incubated for 24 h.100 µL of fresh DMEM medium containing 10 (L 3-(4,5-Dimethylthiazol-2-yl)−2,5-Diphenylte-trazolium Bromide) (MTT) (5 mg/mL) solution was added to each well after removal of the cell medium. The cells were incubated for 4 h. 100 µL DMSO was added to each well after removing the MTT solution. The absorbance at 570 nm was recorded by a microplate reader (Perkin-Elmer Victor3t), and the cells without treatment served as a control. Five replicate measurements were obtained for each sample ($n = 3$).

## Calcein AM/PI assay

HUVECs were cultured in a 96-well plate ($1 \times 10^4$ cells/well) for 24 h, and then added 100 µL different concentrations (0, 1, 2, 5, and 10 µM) of AIE PSs medium for 30 min after the medium was removed. The AIE PSs/HUVECs medium was in the darkness or under white light irradiation for 10 min and then cultured for another 24 h. The culture medium was removed, and the cells were washed with PBS twice, 100 µL Calcein AM/PI solution was added into each well, and the mixture was incubated for 20 min. Calcein AM/PI solution was removed from the dish, and 200 µL PBS was added before observing by CLSM (LSM900, Carl Zeiss, Germany).

## Statistical analyses

The statistical graphs were performed by using OriginPro 8.5.1 and GraphPad Prism 9.3.0. The results are expressed as mean ± standard deviation (SD).

## Reporting summary

Further information on research design is available in the Nature Research Reporting Summary linked to this article.

## Data availability

The authors declare that the source underlying Figs. 2c, d, 4a–f, 5a–d, 6a–h, 8a–d, 8f, g and Supplementary Figs. 13A, B, 17A, B, 18A, B, 19A, B, 20A–F, 21A, B, 26A–D, 33A–D, 39 are provided as a source date file. All other data supporting the finding of this study are available ether in the article and/ or its supplementary information files. The X-ray crystallographic coordinates for structures reported in this study have been deposited at the Cambridge Crystallographic Data Centre (CCDC), under deposition numbers CCDC: 2165387 (https://doi.org/10.5517/ccdc.csd.cc2bp86j) 2165372 (https://doi.org/10.5517/ccdc.csd.cc2bp7q0). Source data are provided with this paper.

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

## Acknowledgements

This work was under financial supports from the National Natural Science Foundation of China (22105016, H.-Q.P.; 22005195, Y.L.), the Open Fund of Guangdong Provincial Key Laboratory of Luminescence from Molecular Aggregates, South China University of Technology (2019B030301003, H.-Q.P.).

## Author contributions

W.Z. synthesized all materials, grew the crystals, performed all photophysical measurements, and wrote the manuscript. Y.L. performed all antifungal/bacterial experiments and revised the manuscript. S.G. and H.L. performed the theoretical calculations. W.-J.G. analyzed the single crystal data. T.P. assisted synthesis. B.L. revised the manuscript. H.-Q.P. and B.Z.T. provided intellectual input and revised the manuscript.

## Competing interests

The authors declare no competing interests.
