## [Peer Review File · Nature Communications]

Stereoisomeric Engineering of Aggregation-Induced Emission Photosensitizers towards Fungal KillingREVIEWER COMMENTS

Reviewer #1 (Remarks to the Author):

In this manuscript, Tang and coworkers reported two supramolecular polymers based on the typical AIE units (tetraphenylethylene) through the host-guest interactions. These supramolecular polymers showed good ROS generation efficiency and low dark toxicity. The potential of supramolecular assembly as a convenient approach to optimize PDT effects was proposed and confirmed here, which is interesting and should be the highlight of this work. My concern is the self-assembly modes of the two molecules and CB[8], especially for (Z)-TPE-EPy. The formation of the supramolecular polymers should be paid more attention to. In my opinion, a major revision is needed. Addressing the following points may be helpful in improving the quality of this manuscript.

1. How does the author exclude the possibilities of the formation of dimers, tetramers or others for the (Z)-TPE-EPy? The NMR, hydrodynamic diameter and morphological changes could not give direct evidence for the self-assembled structures.
2. The SEM image of (Z)-TPE-EPy@CB[8] in Figure S12C is quite like the morphology of CB[8] itself. A control experiment is needed.
3. No obvious change in the UV-vis spectra was observed after the addition of CB[8] into the (Z)-TPE-EPy. This must be explained.
4. The chemical shift changes are not obvious in Figure 2. In addition, all the peaks either stay at their original places or move to higher fields. This is at odds with the shielding effects of the CB[8].
5. An ICT measurement may help to study the binding behavior.
6. The equivalents of the CB[8] seem arbitrary. 2 equiv in NMR titration, 3 equiv in the UV and FLU study, 4 equiv in the ROS study. Please explain this.
7. The binding behavior of CB[7] and TPE-EPy could be considered to help understand the binding mode of CB[8] with TPE-EPy. Of course, the experiments are not necessarily needed in the current work.
8. The photograph of (Z)-TPE-EPy in aqueous solution and EtOH/H₂O mixtures (v:v, 99:1) taken under irradiation at 365 nm should be added to Figure 1d.
9. An obvious typographical error was found on Page 7.
10. On page 4, Figure S10b should be Figure 10a.
11. Abbreviations and full names of the chemicals should be unified.

Reviewer #2 (Remarks to the Author):

In this contribution an interesting approach is reported to use photosensitizers based on aggregation induced emission compounds to treat fungus with PDT. An AIEgen is developed with two configurations (E and Z) and with pyridinium units that allow complexation with CB. The investigations reports a detailed analysis after the effect of the configuration of type of ROS production, and some interesting differences are observed with the E isomer outperforming the Z one. These differences are less clear when the CB complexes are formed. In combination with fungi, the Z isomer

outperforms the E one because of stronger interaction with the fungus. The study provides new insights in the possibility of molecular engineering of AIEgens. Although the authors put much emphasis on the stereomeric effect of ROS production, in practice it is the CB complexation that provides a more clear-cut effect. I would therefore also refocus the story more to the CB complex formation, which strongly diminishes dark toxicity of the components and therefore makes PDT a much more selective process.

Besides this conceptual comment, there are a number of technical comments to be answered.

First of all, more information should be provided on the assembly state of the PSs with CB. From DLS the assemblies seem to be well-defined. This could be corroborated by the PDI values. Furthermore, the EM pictures show crystals rather than the molecular assemblies themselves. AFM would be an appropriate technique to provide this information.

Second, I am surprised by the fact that CB associated PSs also have mitochondrial targeting capability as the pyridinium ions should be shielded by the CB cavity. In fact this shielding is shown by the diminished dark toxicity. These results seem to contradict each other and should be further clarified.

Third the authors have checked the toxicity of their compounds against HUVECs. The conclusion that mammalian cells therefore are not affected is too general. The authors should also look at toxicity toward red blood cells. Furthermore, they don't mention possible toxic effects against bacteria. Also a control experiment in this direction should be performed.

Minor comments

Fig 5 why use 4 equivalents of CB?

Fig S13 please indicate the concentrations (steps) connected to the different curves

Page 7 The paragraph starting with "These results are" is mixed up. Please correct.

REVIEWERS' COMMENTS:

Reviewer #1 (Remarks to the Author):

In this manuscript, Tang and coworkers reported two supramolecular polymers based on the typical AIE units (tetraphenylethylene) through the host-guest interactions. These supramolecular polymers showed good ROS generation efficiency and low dark toxicity. The potential of supramolecular assembly as a convenient approach to optimize PDT effects was proposed and confirmed here, which is interesting and should be the highlight of this work. My concern is the self-assembly modes of the two molecules and CB[8], especially for (Z)-TPE-EPy. The formation of the supramolecular polymers should be paid more attention to. In my opinion, a major revision is needed. Addressing the following points may be helpful in improving the quality of this manuscript.

1. How does the author exclude the possibilities of the formation of dimers, tetramers or others for the (Z)-TPE-EPy? The NMR, hydrodynamic diameter and morphological changes could not give direct evidence for the self-assembled structures.
2. The SEM image of (Z)-TPE-EPy@CB[8] in Figure S12C is quite like the morphology of CB[8] itself. A control experiment is needed.
3. No obvious change in the UV-vis spectra was observed after the addition of CB[8] into the (Z)-TPE-EPy. This must be explained.
4. The chemical shift changes are not obvious in Figure 2. In addition, all the peaks either stay at their original places or move to higher fields. This is at odds with the shielding effects of the CB[8].
5. An ICT measurement may help to study the binding behavior.
6. The equivalents of the CB[8] seem arbitrary. 2 equiv in NMR titration, 3 equiv in the UV and FLU study, 4 equiv in the ROS study. Please explain this.
7. The binding behavior of CB[7] and TPE-EPy could be considered to help understand the binding mode of CB[8] with TPE-EPy. Of course, the experiments are not necessarily needed in the current work.
8. The photograph of (Z)-TPE-EPy in aqueous solution and EtOH/H₂O mixtures (v:v, 99:1) taken under irradiation at 365 nm should be added to Figure 1d.
9. An obvious typographical error was found on Page 7.
10. On page 4, Figure S10b should be Figure 10a.
11. Abbreviations and full names of the chemicals should be unified.

Reviewer #2 (Remarks to the Author):

In this contribution an interesting approach is reported to use photosensitizers based on aggregation induced emission compounds to treat fungus with PDT. An AIEgen is developed with two configurations (E and Z) and with pyridinium units that allow complexation with CB. The investigations reports a detailed analysis after the effect of the configuration of type of ROS production, and some interesting differences are observed with the E isomer outperforming the Z one. These differences are less clear when the CB complexes are formed. In combination with fungi, the Z isomer outperforms the E one because of stronger interaction

with the fungus.

The study provides new insights in the possibility of molecular engineering of AIEgens. Although the authors put much emphasis on the stereomeric effect of ROS production, in practice it is the CB complexation that provides a more clear-cut effect. I would therefore also refocus the story more to the CB complex formation, which strongly diminishes dark toxicity of the components and therefore makes PDT a much more selective process.

Besides this conceptual comment, there are a number of technical comments to be answered.

First of all, more information should be provided on the assembly state of the PSs with CB. From DLS the assemblies seem to be well-defined. This could be corroborated by the PDI values. Furthermore, the EM pictures show crystals rather than the molecular assemblies themselves. AFM would be an appropriate technique to provide this information.

Second, I am surprised by the fact that CB associated PSs also have mitochondrial targeting capability as the pyridinium ions should be shielded by the CB cavity. In fact this shielding is shown by the diminished dark toxicity. These results seem to contradict each other and should be further clarified.

Third the authors have checked the toxicity of their compounds against HUVECs. The conclusion that mammalian cells therefore are not affected is too general. The authors should also look at toxicity toward red blood cells. Furthermore, they don't mention possible toxic effects against bacteria. Also a control experiment in this direction should be performed.

Minor comments

Fig 5 why use 4 equivalents of CB?

Fig S13 please indicate the concentrations (steps) connected to the different curves

Page 7 The paragraph starting with "These results are" is mixed up. Please correct.

Point-by-point response to the reviewers' comments:

Reviewer #1

In this manuscript, Tang and coworkers reported two supramolecular polymers based on the typical AIE units (tetraphenylethylene) through the host-guest interactions. These supramolecular polymers showed good ROS generation efficiency and low dark toxicity. The potential of supramolecular assembly as a convenient approach to optimize PDT effects was proposed and confirmed here, which is interesting and should be the highlight of this work. My concern is the self-assembly modes of the two molecules and CB[8], especially for (Z)-TPE-EPy. The formation of the supramolecular polymers should be paid more attention to. In my opinion, A major revision is needed. Addressing the following points may be helpful in improving the quality of this manuscript.

Response: We would like to thank the reviewer for his/her positive comments and valuable suggestions, which are greatly helpful for improving our manuscript. In this revision, we have added more characterizations (i.e., atomic force microscopy and isothermal titration calorimetry experiments) to explain the self-assembly modes of the stereoisomers and CB[8] (*vide infra*). The details and point-by-point responses are listed as follows:

Comment 1: *How does the author exclude the possibilities of the formation of dimers, tetramers or others for the (Z)-TPE-EPy? The NMR, hydrodynamic diameter and morphological changes could not give direct evidence for the self-assembled structures.*

Response: Thank you for this insightful comment. According to the NMR (Figure 2) and ITC (Figure S12) results, (Z)-TPE-EPy could enter into CB[8] with a binding stoichiometry of 1:1. In addition, the hydrodynamic diameter (Figure 3a) and morphological analysis (Figure S13 and S14) indicated the existence of large-sized supramolecular assemblies in the (Z)-TPE-EPy@CB[8] solution. For these reasons, the formation of supramolecular polymers was inferred. However, we cannot exclude the possibilities of the formation of dimers and other oligomers since the concentrations of the monomers we used in this work were relatively low. Toward a more precise description of the self-assembled structures, “supramolecular polymers” was replaced by “supramolecular assemblies” in the revised manuscript.

Comment 2: *The SEM image of (Z)-TPE-EPy@CB[8] in Figure S12C is quite like the morphology of CB[8] itself. A control experiment is needed.*

Response: Thank you for the valuable suggestion. We have updated Figure S13 that contains the image of the disordered morphology of CB[8] *per se* for comparison. It is totally different from the morphology of (Z)-TPE-EPy@CB[8].

Supplementary Figure S13. SEM images of A) CB[8], B) (Z)-TPE-EPy, C) (Z)-TPE-EPy@CB[8], D) (E)-TPE-EPy and E) (E)-TPE-EPy@CB[8]. The concentration of each sample was 10 μM and the scale bar was 1 μm .

Comment 3: No obvious change in the UV-vis spectra was observed after the addition of CB[8] into the (Z)-TPE-EPy. This must be explained.

Response: We carried out the UV-vis titration experiments of (Z)-TPE-EPy with CB[8]. As shown in Figure S15, (Z)-TPE-EPy exhibited a broad absorption peaked at 376 nm. With the gradual addition of CB[8] from 0 to 4 equiv, the absorption peak was decreased and narrowed, which could be due to the restriction of rotation of the styrylpyridine units in (Z)-TPE-EPy@CB[8]. However, the changes of the UV-vis spectra were not obvious compared to those of the (E)-isomer with CB[8] titration, presumably as a result of the steric hindrance from the V-shaped (Z)-configuration, which hampered the host-enhanced π - π interaction of the end groups on two separate (Z)-TPE-EPy molecules. This presumption can be corroborated by the lower binding constant of (Z)-TPE-EPy with CB[8]. We added further explanation and discussion on the UV-vis changes of the (Z)-isomer with CB[8] titration in the revised manuscript and Supporting Information as follows:

Highlighted in red color on page 7 of the revised manuscript:

“(Z)-TPE-EPy exhibits a broad absorption peaked at 376 nm. With the gradual addition of CB[8] from 0 to 4 equiv, the absorption peak was decreased and narrowed, which could be due to the restriction of rotation of the styrylpyridine units on (Z)-TPE-EPy@CB[8] (Figure 3b and S15A).”

In the caption of Figure S15 in the revised Supporting Information:

Supplementary Figure S15. A) UV-vis absorption titration spectra of (*Z*)-TPE-EPy (30 μM) with different amounts of CB[8] in ultrapure water. Insets: Plots of the absorbance at 376 nm. B) UV-vis absorption titration spectra of (*E*)-TPE-EPy (30 μM) with different amounts of CB[8] in ultrapure water. Insets: Plots of the absorbance intensity at 402 or 462 nm. The changes of the UV-vis spectra of (*Z*)-TPE-EPy were not obvious compared to those of the (*E*)-isomer with CB[8] titration, presumably as a result of the steric hindrance from the V-shaped (*Z*)-configuration, which hampered the host-enhanced π - π interaction of the end groups on two separate (*Z*)-TPE-EPy molecules.

Comment 4: The chemical shift changes are not obvious in Figure 2. In addition, all the peaks either stay at their original places or move to higher fields. This is at odds with the shielding effects of the CB[8].

Response: Thanks for the valuable comments. In the previous ^1H NMR titration experiment, we added a small fraction of $\text{DMSO-}d_6$ to D_2O ($\text{DMSO-}d_6/\text{D}_2\text{O}$, v/v , 1/5) to increase the solubility of the stereoisomers. It was unexpected that $\text{DMSO-}d_6$ had a significant influence on the chemical shift changes in NMR spectra. We are deeply sorry for causing the confusions. We redid the ^1H NMR titration experiment in pure D_2O and updated Figure 2 in the revised manuscript. All the corresponding discussions were revised as follows:

Highlighted in red color on page 5 and 6 of the revised manuscript:

“The specific proton chemical shifts and splitting of the stereoisomers resemble each other (Figure 2). With the titration of CB[8] from 0 to 4.0 equiv, the H_a - H_f proton resonances of (*Z*)-TPE-EPy shift upfield by 0.33~1.07 ppm, indicating their encapsulation in CB[8] cavity with shielding effect. In contrast, H_g - H_i on the (*Z*)-isomer shift downfield by 0.09~0.47 ppm, revealing that their locations are outside but near the carbonyl-rimmed portal of CB[8] with deshielding effect. Similarly, when (*E*)-TPE-EPy is mixed with CB[8] in D_2O (0 to 4.0 equiv), its protons H_a - H_f show remarkable upfield shifts from 0.26 to 0.88 ppm. This demonstrates that the 4-styrylpyridine units are wrapped deeply in the CB[8] cavity. The protons H_g - H_i downfield shift by 0.17 ppm, which should be attributed to their locations on or near the CB[8] portal. Meanwhile, ^1H NMR spectra of (*Z*)- and (*E*)-TPE-EPy@CB[8] displayed pronounced splitting peaks of CB[8] protons, particularly H_x situated toward the interior of the cavity. This suggests an asymmetric charge density environment for two CB[8] portals, demonstrating the host-guest recognitions of CB[8] with (*Z*)- and (*E*)-TPE-EPy.”

Figure 2. Verification of supramolecular assembly by ^1H NMR titration experiment. ^1H NMR titration spectra of (a) (Z)-TPE-EPy and (b) (E)-TPE-EPy with different amounts of CB[8] in D_2O . $[(Z)\text{-TPE-EPy}] = [(E)\text{-TPE-EPy}] = 0.25 \text{ mM}$.

Comment 5: An ICT measurement may help to study the binding behavior.

Response: We believe that the reviewer is referring to isothermal titration calorimetry (ITC), which is an important approach to study the thermodynamic information of host-guest binding. According to this nice advice, we carried out the ITC titration experiments of the stereoisomers and CB[8]. The results suggest a binding stoichiometry of 1:1 for (Z)-TPE-EPy@CB[8] and (E)-TPE-EPy@CB[8]. The new data have been added as Figure S12. The corresponding discussion was added in the revised manuscript as follows:

Highlighted in red color on page 6 of the revised manuscript:

“Additionally, the binding behaviors between the stereoisomers and CB[8] were also estimated by isothermal titration calorimetry (ITC) experiments, which indicated a stoichiometry of 1:1 for (Z)- and (E)-TPE-EPy@CB[8] assemblies (Figure S12).”

Supplementary Figure S12. ITC data for (Z)-TPE-EPy and (E)-TPE-EPy with CB[8] in H_2O . $[(Z)\text{-TPE-EPy}] = [(E)\text{-TPE-EPy}] (\text{cell}) = 50 \mu\text{M}$, $[\text{CB}[8]] (\text{syringe}) = 350 \mu\text{M}$, 298 K.

Comment 6: The equivalents of the CB[8] seem arbitrary. 2 equiv in NMR titration, 3 equiv in the UV and FLU study, 4 equiv in the ROS study. Please explain this.

Response: The experimental results demonstrate that the NMR, UV and PL spectra, and ROS generation of the supramolecular assemblies could approach a plateau at 2, 3, and 4 equiv CB[8], respectively. (Figure 2, S15, S16, and S19). We apologize for the confusion caused by these inconsistent equivalents. To avoid confusing the readers, the equivalent of CB[8] was unified to 4 equiv. in different studies. The corresponding figures and contents have been updated in the revised manuscript as follows:

Highlighted in red color on page 5-6 of the revised manuscript:

“With the titration of CB[8] from 0 to 4.0 equiv, the H_a-H_f protons of (Z)-TPE-EPy shift upfield by 0.33~1.07 ppm, indicating their encapsulation in CB[8] cavity with shielding effect.”

“Similarly, when (E)-TPE-EPy is mixed with CB[8] in D₂O (0 to 4.0 equiv), its protons H_a-H_f show remarkable upfield shifts from 0.26 to 0.88 ppm.”

Figure 2. Verification of supramolecular assembly by ¹H NMR titration experiment. ¹H NMR titration spectra of (a) (Z)-TPE-EPy and (b) (E)-TPE-EPy with different amounts of CB[8] in D₂O. [(Z)-TPE-EPy]=[E)-TPE-EPy]=0.25 mM.

Highlighted in red color on page 6-7 of the revised manuscript:

“With the gradual addition of CB[8] from 0 to 4 equiv, the absorption peak was decreased and narrowed, which could be due to the restriction of rotation of the styrylpyridine units in (Z)-TPE-EPy@CB[8].”

“By comparison, with the CB[8] (from 0 to 4 equiv) gradually adding into the (E)-TPE-EPy aqueous solution, the absorption of (E)-TPE-EPy decreases and also red-shift with the two absorbance maxima shifting from 316 and 402 nm to 338 and 462 nm, accompanying by a visible solution color change from bright green to yellow (Figure 3c and Figure S15B)”

Supplementary Figure S15. A) UV-vis absorption titration spectra of (*Z*)-TPE-EPy (30 μ M) with different amounts of CB[8] in ultrapure water. Insets: Plots of the absorbance at 376 nm. B) UV-vis absorption titration spectra of (*E*)-TPE-EPy (30 μ M) with different amounts of CB[8] in ultrapure water. Insets: Plots of the absorbance intensity at 402 or 462 nm. The changes of the UV-vis spectra of (*Z*)-TPE-EPy were not obvious compared to those of the (*E*)-isomer with CB[8] titration, presumably as a result of the steric hindrance from the V-shaped (*Z*)-configuration, which hampered the host-enhanced π - π interaction of the end groups on two separate (*Z*)-TPE-EPy molecules.

Supplementary Figure S16. A) Fluorescence titration spectra of (*Z*)-TPE-EPy (30 μ M) with different amounts of CB[8] (from 0 to 4 equiv) in ultrapure water. B) Fluorescence titration spectra of (*E*)-TPE-EPy (30 μ M) with different amounts of CB[8] (from 0 to 4 equiv) in ultrapure water. The linear fitting curves of fluorescence intensity C) from Figure A) at 638 nm and D) from Figure B) at 686 nm.

Highlighted in red color on page 10 of the revised manuscript:

“As shown in Figure 5a, 5b, and S19, the stereoisomers exhibit improved ROS generation efficiency after they self-assembled with the CB[8] host, which is consistent with the aforementioned results of fluorescence enhancement.”

Figure S19. ROS generation from A) (Z)-TPE-EPy and B) (E)-TPE-EPy with different equivalents of CB[8], upon white light irradiation for 2 minutes, using DCFH (5 μ M) as an indicator.

Comment 7: The binding behavior of CB[7] and TPE-EPy could be considered to help understand the binding mode of CB[8] with TPE-EPy. Of course, the experiments are not necessarily needed in the current work.

Response: Thanks for the valuable comments. We studied the binding behavior of CB[7] and TPE-EPy as shown in Figure S17. In particular, upon the gradual addition of CB[7] into the (E)-TPE-EPy aqueous solution, the UV-vis absorption curves of (E)-TPE-EPy@CB[7] decrease gradually with only a slight red-shift. This can be explained by the small cavity of CB[7] that can encapsulate only one (E)-TPE-EPy end group, thereby excluding the host-enhanced π - π interaction of two separate (E)-isomers. Such a difference between (E)-TPE-EPy@CB[7] and (E)-TPE-EPy@CB[8] corroborates the binding mode of CB[8] with the (E)-isomer as we described in the manuscript. We have added a new figure in the Supporting Information (Figure S17) and new descriptions in the revised manuscript as follows for better understanding the self-assembly behaviors of CB[8] and TPE-EPy:

Highlighted in red color on page 8 of the revised manuscript:

“This is why CB[7] with a smaller cavity that encapsulates only one (E)-isomer end group can exclude the pronounced red-shift absorption (Figure S17).”

Figure S17. A) UV-vis absorption titration spectra of (Z)-TPE-EPy (30 μM) with different amounts of CB[7] in ultrapure water. B) UV-vis absorption titration spectra of (E)-TPE-EPy (30 μM) with different amounts of CB[8] in ultrapure water.

Comment 8: The photograph of (Z)-TPE-EPy in aqueous solution and EtOH/H₂O mixtures (v:v, 99:1) taken under irradiation at 365 nm should be added to Figure 1d.

Response: This photograph has been added in Figure 1d.

Figure 1d. Plot of relative emission intensity versus the EtOH fraction (vol %) of (Z)- and (E)-TPE-EPy in H₂O/EtOH mixtures (10 μM , λ_{em} =645 nm). Inset: photographs of (Z)- and (E)-TPE-EPy in aqueous solution and EtOH/H₂O mixtures (v:v, 99:1) taken under irradiation at 365 nm.

Comment 9: An obvious typographical error was found on Page 7.

Response: We apologize for the typographical error. The corresponding part has been corrected in the revised manuscript and highlighted in red color as shown on Page 7.

“These results are consistent with the AIE characteristics of the stereoisomers. Detailly, the intramolecular motions of (Z)-TPE-EPy are restricted by host-guest interactions, which

suppress the nonradiative energy dissipation and lead to stronger fluorescence signals. By comparison, with the CB[8] (from 0 to 4 equiv) gradually adding into the (*E*)-TPE-EPy aqueous solution, the absorption of (*E*)-TPE-EPy decreases and also red-shift with the two absorbance maxima shifting from 316 and 402 nm to 338 and 462 nm, accompanying by a visible solution color change from bright green to yellow (Figure 3c and Figure S15B).”

Comment 10: On page 4, Figure S10b should be Figure 10a.

Response: Thank you for pointing out this mistake. “Figure S10b” has been corrected as “Figure S10A” on page 4.

Comment 11: Abbreviations and full names of the chemicals should be unified.

Response: The abbreviations and full names of chemicals, including triethylamine (TEA), dichloromethane (DCM), *N, N*-Dimethylformamide (DMF), and tetrahydrofuran (THF) have been unified.

Reviewer #2

In this contribution an interesting approach is reported to use photosensitizers based on aggregation induced emission compounds to treat fungus with PDT. An AIEgen is developed with two configurations (E and Z) and with pyridinium units that allow complexation with CB. The investigations reports a detailed analysis after the effect of the configuration of type of ROS production, and some interesting differences are observed with the E isomer outperforming the Z one. These differences are less clear when the CB complexes are formed. In combination with fungi, the Z isomer outperforms the E one because of stronger interaction with the fungus.

The study provides new insights in the possibility of molecular engineering of AIEgens. Although the authors put much emphasis on the stereomeric effect of ROS production, in practice it is the CB complexation that provides a more clear-cut effect. I would therefore also refocus the story more to the CB complex formation, which strongly diminishes dark toxicity of the components and therefore makes PDT a much more selective process.

Response: We would like to thank you for your positive comments on the stereoisomeric engineering of aggregation-induced emission photosensitizers and valuable suggestions to improve our manuscript. After providing more direct evidences, such as atomic force microscopy (AFM) images, isothermal titration calorimetry (ITC) experiments, and analysis of the host-guest effect of CB[8] on mitochondrial targeting capability based on the reviewer's suggestions, the significance of the CB[8] complex formation was further emphasized in the revised manuscript.

Besides this conceptual comment, there are a number of technical comments to be answered.

First of all, more information should be provided on the assembly state of the PSs with CB. From DLS the assemblies seem to be well-defined. This could be corroborated by the PDI values. Furthermore, the EM pictures show crystals rather than the molecular assemblies themselves. AFM would be an appropriate technique to provide this information.

Response: Thank you for your valuable suggestions. The PDI values have been added in the caption of Figure 3a, which are 0.189 for (Z)-TPE-EPy, 0.141 for (Z)-TPE-EPy@CB[8], 0.195 for (E)-TPE-EPy, and 0.177 for (E)-TPE-EPy@CB[8]. In addition, we added AFM images of the stereoisomers and their corresponding supramolecular assemblies in the Supporting Information (Figure S14). In combination with the SEM pictures, the results corroborate the formation of well-defined morphologies of the supramolecular assemblies.

Supplementary Figure S14. AFM images of A) (Z)-TPE-EPy, B) (E)-TPE-EPy, C) (Z)-TPE-EPy@CB[8], D) (E)-TPE-EPy@CB[8]. The concentration of each sample was 10 μM .

Second, I am surprised by the fact that CB associated PSs also have mitochondrial targeting capability as the pyridinium ions should be shielded by the CB cavity. In fact this shielding is shown by the diminished dark toxicity. These results seem to contradict each other and should be further clarified.

Response: We agree with the reviewer’s comments. The pyridinium cations were shielded by the CB[8] cavity, which resulted in the diminished dark toxicity of the stereoisomers. We believe that such a shielding effect also reduced the mitochondrial targeting capabilities of the PSs since the Pearson’s correlation coefficients between the PSs and Mito-Tracker DR decreased for both (Z)- and (E)-isomers after their binding with CB[8] (Figure 6). However, it was demonstrated that the Zeta potential of the supramolecular assemblies is higher than (Z)- and (E)-TPE-EPy *per se* (Figure S25), presumably because of the shape and size changes of their morphologies. In this regard, the host-guest self-assembly resulted in higher binding efficiency of the supramolecular PSs. For these reasons, CB[8] associated PSs also have mitochondrial targeting capability without apparent decreases. We have added a brief explanation in the revised manuscript as follows:

Highlighted in red color on page 12 of the revised manuscript:

“Meanwhile, the Pearson’s correlation coefficients demonstrate that the shielding effect has no significant effect on mitochondrial targeting capabilities of the supramolecular PSs, presumably due to their higher Zeta potentials resulted from self-assembly of (Z)- and (E)-isomers (Figure S25).”

Third the authors have checked the toxicity of their compounds against HUVECs. The conclusion that mammalian cells therefore are not affected is too general. The authors should also look at toxicity toward red blood cells. Furthermore, they don’t mention possible toxic

effects against bacteria. Also a control experiment in this direction should be performed.

Response: Thanks for your valuable suggestions. We have studied toxicity of the PSs towards red blood cells. As shown in Figure S37, hemolysis analysis in red blood cells indicates that both the stereoisomers and their supramolecular assemblies have good blood compatibility. Additionally, we incubated the PSs with the Gram-positive bacterium *S. aureus* and Gram-negative bacterium *E. coli*. The results show that the bacteria are stained with bright fluorescence, indicating good binding efficiency of the PSs to the bacteria. Moreover, upon light irradiation, the killing efficiency of these PSs to *S. aureus* and *E. coli* can reach almost 100% at the concentration of 0.5 μM and 5 μM , respectively, suggestive of the excellent antibacterial effect of the stereoisomers and their corresponding supramolecular assemblies. All the control experiments have been added in Figure S26-S31 and Figure S37 and the relevant description was added in the revised manuscript as follows:

Highlighted in red color on page 14, right column of the revised manuscript:

“In addition, hemolysis analysis in red blood cells demonstrates that both the stereoisomers and their supramolecular assemblies have good blood compatibility (Figure S37). Such good biocompatibility to HUVECs and red blood cells implies that our AIE PSs are ideal photosensitizers for achieving selective killing of fungus with low side effects in clinical applications.”

Figure S37. Hemolysis activity of different AIE PSs (10 μM), NaCl (0.9% in water) and water.

Highlighted in red color on page 14, left column of the revised manuscript:

“It is worth to mention that our PSs also have good binding efficiency to bacteria and can exhibit surpass 98% and 99% phototoxicity to *Staphylococcus aureus* (*S. aureus*) and *Escherichia coli* (*E. coli*) at the concentration of 0.5 μM and 5 μM , respectively (Figure S26-S31).”

Supplementary Figure S26. Bright-field and fluorescent images of *S. aureus* A) and *E. coli* B) incubated with the different AIEgen (5 μM) for 10 min. (Z)-TPE-EPy, (E)-TPE-EPy, (Z)-TPE-EPy@CB[8]: $\lambda_{\text{ex}} = 405 \text{ nm}$, $\lambda_{\text{em}} = 600\text{-}700$.

Supplementary Figure S27. Photographs of *S. aureus* cultured on agar plate supplemented with different concentration of (Z)-TPE-EPy (A) and (Z)-TPE-EPy@CB[8] (B) in darkness or upon white light irradiation (16 mW cm^{-2}) for 10 min (n=3)

Supplementary Figure S28. Photographs of *S. aureus* cultured on agar plate supplemented with different concentration of (*E*)-TPE-EPy (A) and (*E*)-TPE-EPy@CB[8] (B) in darkness or upon white light irradiation (16 mW cm⁻²) for 10 min (n=3).

Figure S29. Photographs of *E. coli* cultured on agar plate supplemented with different concentration of (*Z*)-TPE-EPy (A) and (*Z*)-TPE-EPy@CB[8] (B) in the darkness or upon white light irradiation (16 mW cm⁻²) for 10 min (n=3).

Figure S30. Photographs of *E.coli* cultured on agar plate supplemented with different concentration of (E)-TPE-EPy (A) and (E)-TPE-EPy@CB[8] (B) in the darkness or upon white light irradiation (16 mW cm^{-2}) for 10 min ($n=3$).

Figure S31. Colony count of A), B) *S. aureus* and C), D) *E. coli* in the darkness or upon light irradiation (16 mW cm^{-2}) following the treatment of different concentrations of (Z)-TPE-EPy, (E)-TPE-EPy, (Z)-TPE-EPy@CB[8], (E)-TPE-EPy@CB[8] for 10 min, respectively. Dashed lines indicate limit of detection.

Minor comments

Fig 5 why use 4 equivalents of CB?

Response: The experimental results demonstrated that ROS generation of the supramolecular assemblies could approach a plateau at 4 equiv. of the CB[8] (Figure S19). This evidence has been added in the revised manuscript and Supporting Information.

Highlighted in red color on page 10, right column of the revised manuscript:

“As shown in Figure 5a, 5b, and S19, the stereoisomers exhibit improved ROS generation efficiency after they self-assembled with the CB[8] host, which is consistent with the aforementioned results of fluorescence enhancement.”

Figure S19. ROS generation from A) (Z)-TPE-EPy and B) (E)-TPE-EPy with different equivalents of CB[8], upon white light irradiation for 2 minutes, using DCFH (5 μ M) as an indicator.

Fig S13 please indicate the concentrations (steps) connected to the different curves

Response: Thank you for pointing out this oversight. The concentrations of the stereoisomers, and CB[8] corresponding to different curves have been added in Figure S16.

Supplementary Figure S16. A) Fluorescence titration spectra of (Z)-TPE-EPy (30 μ M) with different amounts of CB[8] (from 0 to 4 equiv) in ultrapure water. B) Fluorescence titration spectra of (E)-TPE-EPy (30 μ M) with different amounts of CB[8] (from 0 to 4 equiv) in ultrapure water. The linear fitting curves of fluorescence intensity C) from Figure A) at 638 nm and D) from Figure B) at 686 nm.

Page 7 The paragraph starting with “These results are” is mixed up. Please correct.

Response: We are deeply sorry for the typographical error. The corresponding part has been corrected and highlighted in red color as follows:

Highlighted in red color on page 7 of the revised manuscript:

“These results are consistent with the AIE characteristics of the stereoisomers. Detailly, the intramolecular motions of (Z)-TPE-EPy are restricted by host-guest interactions, which suppress the nonradiative energy dissipation and lead to stronger fluorescence signals. By comparison, with the CB[8] (from 0 to 4 equiv) gradually adding into the (E)-TPE-EPy aqueous solution, the absorption of (E)-TPE-EPy decreases and also red-shift with the two absorbance maxima shifting from 316 and 402 nm to 338 and 462 nm, accompanying by a visible solution color change from bright green to yellow (Figure 3c and Figure S15B).”

REVIEWERS' COMMENTS

Reviewer #1 (Remarks to the Author):

The manuscript has been greatly improved. The newly added ITC measurement and the revised NMR titration experiment help a lot to support the author's hypothesis. However, I still worried about the morphology of (Z)-TPE-EPy@CB[8] in Figure S13c. It looks too perfect to me. The complex seems single crystalline. The authors may continue to explore the atomic-level structure of the complex by using X-ray or synchrotron radiation if Figure S13C is true. Anyway, although some open questions remain, I believe the current version could be accepted for now.

Reviewer #2 (Remarks to the Author):

The authors have comprehensively addressed my major comments with additional experiments and explanations. As a result, besides an advice to polish the English somewhat, there are no technical comments or concerns regarding this paper, which has now the novelty and scientific level that fits the journal.

REVIEWERS' COMMENTS:

Reviewer #1 (Remarks to the Author):

The manuscript has been greatly improved. The newly added ITC measurement and the revised NMR titration experiment help a lot to support the author's hypothesis. However, I still worried about the morphology of (Z)-TPE-EPy@CB[8] in Figure S13c. It looks too perfect to me. The complex seems single crystalline. The authors may continue to explore the atomic-level structure of the complex by using X-ray or synchrotron radiation if Figure S13C is true. Anyway, although some open questions remain, I believe the current version could be accepted for now.

Reviewer #2 (Remarks to the Author):

The authors have comprehensively addressed my major comments with additional experiments and explanations. As a result, besides an advice to polish the English somewhat, there are no technical comments or concerns regarding this paper, which has now the novelty and scientific level that fits the journal.

Point-by-point response to the reviewers' comments:

Reviewer #1

The manuscript has been greatly improved. The newly added ITC measurement and the revised NMR titration experiment help a lot to support the author's hypothesis. However, I still worried about the morphology of (Z)-TPE-EPy@CB[8] in Figure S13c. It looks too perfect to me. The complex seems single crystalline. The authors may continue to explore the atomic-level structure of the complex by using X-ray or synchrotron radiation if Figure S13C is true. Anyway, although some open questions remain, I believe the current version could be accepted for now.

Responses: We sincerely thank the reviewer for the comments and supporting publication of our revised manuscript. Thanks again for your valuable suggestions on the exploration of the supramolecular complex structures, which are very helpful for our future work. We will continue to study some CB[8]-related supramolecular assemblies and compare their corresponding morphologies for understanding the formation of these ordered structures.

Reviewer #2

The authors have comprehensively addressed my major comments with additional experiments and explanations. As a result, besides an advice to polish the English somewhat, there are no technical comments or concerns regarding this paper, which has now the novelty and scientific level that fits the journal.

Responses: We sincerely thank the reviewer's recognition of our revised manuscript. We have an international team and some colleagues are native speakers in English. As suggested by the reviewer, our colleagues have carefully polished the English writing in the manuscript.